# Towards Better Optimization For Listwise Preference in Diffusion Models

**Jiamu Bai[1,*], Xin Yu[1,*], Meilong Xu[3], Weitao Lu[2], Xin Pan[2],**
**Kiwan Maeng[1], Daniel Kifer[1], Jian Wang[2], Yu Wang[2]**

[1]Penn State University     [2]TikTok Inc.     [3]Stony Brook University

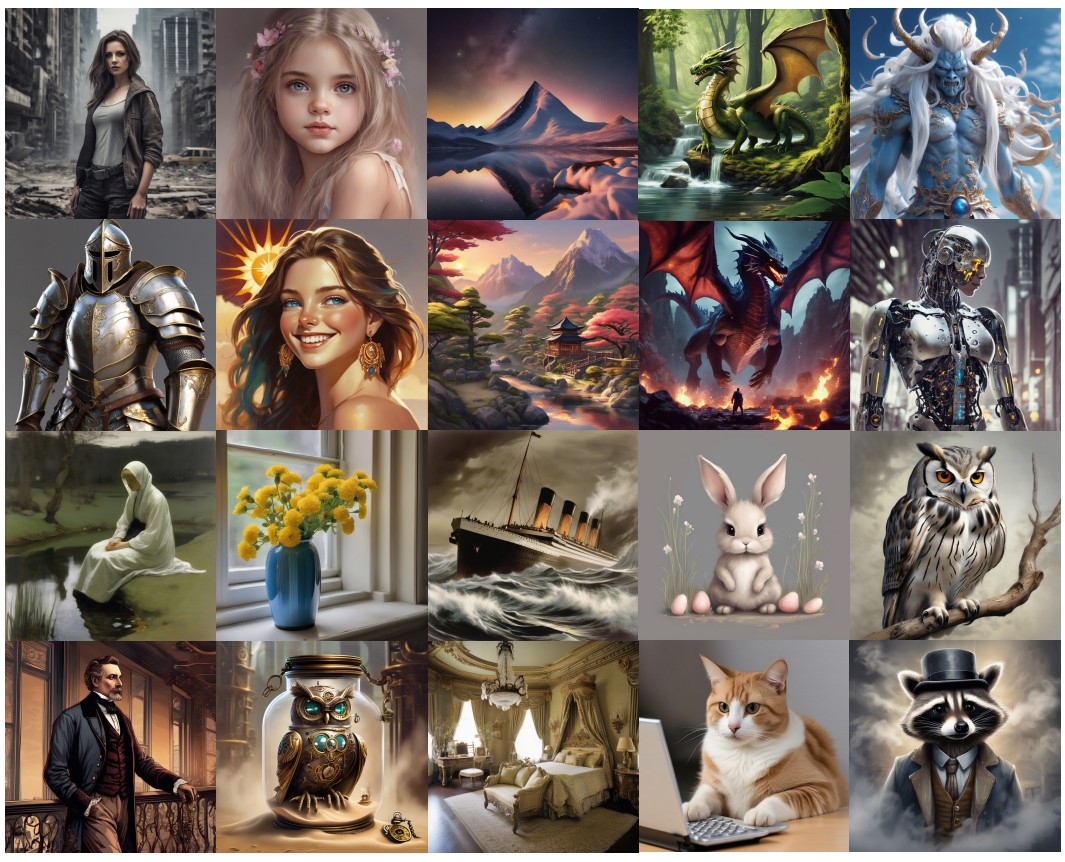

Figure 1: Sample images generated from SDXL trained with Diffusion-LPO. Diffusion-LPO generalizes Diffusion-DPO by optimizing the preference under a list of ranked images. After finetuning with Diffusion-LPO, SDXL produces images with higher visual aesthetics and prompt alignments.

## Abstract

Reinforcement learning from human feedback (RLHF) has proven effectiveness for aligning text-to-image (T2I) diffusion models with human preferences. Although Direct Preference Optimization (DPO) is widely adopted for its computational efficiency and avoidance of explicit reward modeling, its applications to diffusion models have primarily relied on pairwise preferences. The precise optimization of listwise preferences remains largely unaddressed. In practice, human feedback on image preferences often contains implicit ranked information, which

* Co-first author. Work done during Jiamu Bai's internship at TikTok.
1:{jvb6867,xmy5152,kvm6242,duk17}@psu.edu
2:{weitao.lu,xin.pan,jian.wang1,yuwang.w}@bytedance.com
3:meixu@cs.stonybrook.edu

conveys more precise human preferences than pairwise comparisons. In this work, we propose **Diffusion-LPO**, a simple and effective framework for **Listwise Preference Optimization** in diffusion models with listwise data. Given a caption, we aggregate user feedback into a ranked list of images and derive a listwise extension of the DPO objective under the Plackett–Luce model. Diffusion-LPO enforces consistency across the entire ranking by encouraging each sample to be preferred over all of its lower-ranked alternatives. We empirically demonstrate the effectiveness of Diffusion-LPO across various tasks, including text-to-image generation, image editing, and personalized preference alignment. Diffusion-LPO consistently outperforms pairwise DPO baselines on visual quality and preference alignment.

## 1 INTRODUCTION

Diffusion models, including prominent architectures like Stable Diffusion (Podell et al., 2023; Rombach et al., 2022), Imagen (Saharia et al., 2022), and DALL-E 2 (Ramesh et al., 2022), have become a dominant paradigm in text-to-image synthesis due to their ability to generate high-fidelity and semantically aligned images (Ho et al., 2020). While these models are pretrained on massive web-scale datasets to establish a powerful foundation, real-world deployment necessitates further finetuning to align their outputs more closely with human preferences. Drawing inspiration from alignment techniques in large language models (LLMs), recent works have adapted methods like Direct Preference Optimization (DPO) for diffusion models, enabling preference learning without an explicit reward model (Yang et al., 2024; Wallace et al., 2024; Zhu et al., 2025; Li et al., 2024; Gu et al., 2024). These approaches have demonstrated superior performance over standard supervised fine-tuning by directly incorporating preference signals into the optimization process.

Despite these advances, prior works mostly rely on pairwise human preference data. Pairwise comparisons are relatively easy to collect and annotate, making them the dominant form in existing human preference datasets (Bai et al., 2022; Kirstain et al., 2023). However, human preferences are inherently expressed as ranked lists rather than isolated pairs, since

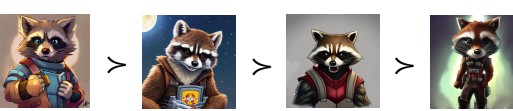

Figure 2: An example of a ranked list of images under human preference. The caption is "Rocket Raccoon, furry art, fanart, digital painting."

preferences are not simply absolute approvals or rejections but relative orderings among multiple options, as shown in the example of Figure 2. In this sense, pairwise annotations implicitly contain richer ranking information: if we consider a user indicating their preferences as response $\mathbf{x}^{(a)} \succ \mathbf{x}^{(b)}$ and $\mathbf{x}^{(b)} \succ \mathbf{x}^{(c)}$, this naturally implies the transitive ordering $\mathbf{x}^{(a)} \succ \mathbf{x}^{(b)} \succ \mathbf{x}^{(c)}$, which more faithfully represents the user's overall preference. Notably, we observe that in the Pick-a-Pic dataset (Kirstain et al., 2023), where each user provides pairwise preferences for images, 56% of such annotations can be benefited from grouping into consistent rankings with size greater than 2 to form more informative human preferences. This motivates the need for listwise optimization, which directly models the ranking nature of human preferences. A straightforward approach to formulating an objective for listwise preference is to decompose rankings of size $m$ into $m(m-1)/2$ pairs to apply the pairwise DPO objective. Several works (Chen et al., 2025; Karthik et al., 2024) have explored this direction, but their methods rely on auxiliary evaluators to assign reward scores to each image, introducing extra computational overhead and resource burden.

In this work, we introduce Diffusion-LPO (Listwise Preference Optimization), a new framework for aligning diffusion models with user-level preferences. Instead of reducing user feedback to pairwise comparisons, Diffusion-LPO directly models listwise preferences with the Plackett-Luce ranking model (Plackett, 1975). Unlike pairwise DPO, which only compares a winning sample against a single loser, Diffusion-LPO enforces consistency across the entire ranking by encouraging each sample to be preferred over all of its lower-ranked alternatives, thereby preserving the full relative order within the list. When the list size equals 2, Diffusion-LPO degenerates to Diffusion-DPO. As a general approach for listwise preference optimization, the formulation of Diffusion-LPO can potentially enhance many existing methods built upon the Diffusion-DPO family.

To evaluate the effectiveness of our method, we train SD1.5 (Rombach et al., 2022) and SDXL (Podell et al., 2023) on the Pick-a-Pic dataset (Kirstain et al., 2023), a dataset of real-world user preferences that contains human preference pairs of images. We uncover the inherent listwise preference within the dataset, with detailed information of dataset reformulation illustrated in Section 4.1. Empirical results show that Diffusion-LPO improves PickScore win rates by 12% and by 4% over Diffusion-DPO on SD1.5 and SDXL. Specifically, our contribution can be listed as follows:

- We first provide a novel perspective that ranked human preferences are implicitly embedded within pairwise annotations. It motivates us to intentionally enforce the diffusion model to learn such structural rank information.

- We propose Diffusion-LPO, a direct preference optimization framework from listwise human feedback. As a generalization of pairwise DPO, Diffusion-LPO can be integrated into any DPO-based method to further boost the performance.

- We empirically demonstrate the effectiveness of Diffusion-LPO across various T2I tasks. Diffusion-LPO exhibits higher alignment with human preferences over baselines.

## 2 RELATED WORK

**RLHF Alignment**    Reinforcement Learning from Human Feedback (RLHF) has emerged as the dominant paradigm for aligning large language models (LLMs) with human preferences (Christiano et al., 2017; Stiennon et al., 2020; Mnih et al., 2016; Ziegler et al., 2019; Bai et al., 2022; Bakker et al., 2022; Jiang et al., 2025). In its standard formulation, a reward model is trained to approximate user preferences, and the policy is subsequently optimized via reinforcement learning with this reward signal. While effective, this two-stage pipeline is computationally expensive and can be susceptible to reward hacking (Singhal et al., 2023; Chen et al., 2024). To address these limitations, recent work has explored direct preference optimization (DPO) (Rafailov et al., 2023), which bypasses explicit reward modeling and instead optimizes the policy to prefer human-preferred outputs directly. Most existing approaches rely on pairwise preference data and the Bradley–Terry (BT) model (Bradley & Terry, 1952) to parameterize the likelihood of a winning sample over a losing one (Ethayarajh et al., 2024; Meng et al., 2024). However, pairwise comparisons may provide limited information about human preferences, as they reduce complex judgments into binary wins and losses. Several recent works have extended DPO to the listwise ranking setting  (Liu et al., 2025b; Song et al., 2024; Hong et al., 2024a), enabling richer supervision signals from ranked lists of responses than pairs.

**Preference Alignment with Diffusion Models**    Recent work has shown that reinforcement learning is an effective tool for aligning diffusion models with human aesthetic and semantic preferences (Fan et al., 2023; Black et al., 2023; Xue et al., 2025; Clark et al., 2023; Zhao et al., 2025; Uehara et al., 2024; Prabhudesai et al., 2023; Lee et al., 2023; Eyring et al., 2024; Sun et al., 2025a; Lee et al., 2025; Zhang et al., 2025). Among existing reinforcement learning optimization approaches, Direct Preference Optimization (DPO) stands out as an attractive option by defining an implicit reward through preference comparisons, eliminating the need for explicit reward models (Wallace et al., 2024; Yang et al., 2024; Li et al., 2024; Gu et al., 2024; Zhu et al., 2025; Hong et al., 2024b; Lee et al., 2025; Wang et al., 2025; Liang et al., 2024; Sun et al., 2025b; Lu et al., 2025; Wu et al., 2025; Cai et al., 2025; Hu et al., 2025; Zhang et al., 2024). Diffusion-DPO (Wallace et al., 2024) adapts this framework to the diffusion setting. Follow-up work includes DSPO (Zhu et al., 2025) which derives the DPO objective through score matching, Diffusion-KTO (Li et al., 2024) which formulates the objective by maximizing human utility, MaPO (Hong et al., 2024b) which maximizes likelihood margin between preferred and dispreferred image sets. Despite these advances, the above DPO-based methods for diffusion rely exclusively on *pairwise* preference data, while human feedback can come as a list of rankings.

## 3 PRELIMINARIES

### 3.1 DIFFUSION MODELS

Diffusion models are generative models that learn data distributions by reversing a gradual noising process (Ho et al., 2020; Song et al., 2021). The forward (diffusion) process starts from clean data $x_0 \sim p_{\text{data}}$, where $p_{\text{data}}$ is the data distribution, and progressively adds Gaussian noise across $T$ time steps with noise scheduling $\beta_1, ..., \beta_T$:

$$q(\mathbf{x}_t \mid \mathbf{x}_0) = \mathcal{N}\left(\sqrt{\bar{\alpha}_t}\mathbf{x}_0, (1 - \bar{\alpha}_t)I\right), \quad \bar{\alpha}_t = \prod_{s=1}^{t} \alpha_s, \ \alpha_t = 1 - \beta_t.$$

The reverse process (denoising) is parameterized by a neural network $\epsilon_\theta$, which predicts noise $\epsilon_t$ given a noisy image $\mathbf{x}_t$ at timestep $t$. The standard training objective is a weighted denoising score matching loss (Ho et al., 2020):

$$\mathcal{L}_{\text{DM}}(\theta) = \mathbb{E}_{\mathbf{x}_t^{(1:m)} \sim p_\theta(\mathbf{x}_t^{(1:m)} \mid \mathbf{x}_0^{(1:m)})}\left[\omega(\lambda_t) \left\|\epsilon - \epsilon_\theta(\mathbf{x}_t, t)\right\|_2^2\right],$$

where $\lambda_t = \log(\alpha_t^2/\sigma_t^2)$ is the signal-to-noise ratio, $p_\theta$ is the reverse process, and $\omega(\lambda_t)$ a pre-defined weighting function.

### 3.2 DIRECT PREFERENCE OPTIMIZATION (DPO)

Large generative models can be aligned with human feedback via reinforcement learning from human feedback (RLHF) (Christiano et al., 2017; Stiennon et al., 2020). The goal of RLHF is to optimize the policy $\pi_\theta$ by maximizing the reward $r(\mathbf{c}, \mathbf{x}_0)$ given the prompt $\mathbf{c}$ sampled from prompt set $\mathcal{D}_\mathbf{c}$ and sample $\mathbf{x}_0$, with a regularization KL-divergence from a reference policy $\pi_{\text{ref}}(\mathbf{x}_0|\mathbf{c})$:

$$\max_{\pi_\theta} \mathbb{E}_{\mathbf{c} \sim D_\mathbf{c}, \mathbf{x}_0 \sim \pi_\theta(\mathbf{x}_0|\mathbf{c})}[r(\mathbf{c}, \mathbf{x}_0)] - \beta \mathbb{D}_{\text{KL}}[\pi_\theta(\mathbf{x}_0|\mathbf{c})||\pi_{\text{ref}}(\mathbf{x}_0|\mathbf{c})]. \tag{1}$$

Classical RLHF requires training a separate reward model and then optimizing the generative policy with reinforcement learning (e.g., PPO (Schulman et al., 2017)). Direct Preference Optimization (DPO) (Rafailov et al., 2023) eliminates the need for explicit reward modeling by modeling human preferences with the Bradley-Terry (BT) model (Bradley & Terry, 1952):

$$p_{BT}(\mathbf{x}_0^+ \succ \mathbf{x}_0^- \mid \mathbf{c}) = \sigma\left(r(\mathbf{c}, \mathbf{x}_0^+) - r(\mathbf{c}, \mathbf{x}_0^-)\right),$$

where $\mathbf{c}$ is the conditioning prompts and $(\mathbf{x}_0^+, \mathbf{x}_0^-)$ is the pair of human-labeled winning and losing samples. Here, $r(\mathbf{c}, \mathbf{x}_0)$ is the latent reward, and $\sigma$ is the sigmoid function. Use the log-likelihood ratios to get the implicit rewards, the DPO loss becomes

$$\mathcal{L}_{\text{DPO}}(\theta) = -\mathbb{E}_{(\mathbf{c}, \mathbf{x}_0^+, \mathbf{x}_0^-) \sim \mathcal{D}}\left[\log \sigma\left(\beta\left(\log \frac{\pi_\theta(\mathbf{x}_0^+|\mathbf{c})}{\pi_{\text{ref}}(\mathbf{x}_0^+|\mathbf{c})} - \log \frac{\pi_\theta(\mathbf{x}_0^-|\mathbf{c})}{\pi_{\text{ref}}(\mathbf{x}_0^-|\mathbf{c})}\right)\right)\right].$$

Here, $(\mathbf{c}, \mathbf{x}_0^+, \mathbf{x}_0^-) \sim \mathcal{D}$ indicates $\mathbf{c} \sim D_\mathbf{c}, \mathbf{x}_0^+ \sim \pi_\theta(\mathbf{x}_0^+|\mathbf{c}), \mathbf{x}_0^- \sim \pi_\theta(\mathbf{x}_0^-|\mathbf{c})$.

### 3.3 DIFFUSION-DPO

The key observation of adapting DPO to diffusion models is that the diffusion denoising objective $\mathcal{L}_{\text{DM}}$ serves as an implicit reward. Specifically, Diffusion-DPO (Wallace et al., 2024) defines a reward for an image $x$ at timestep $t$ as the improvement of the fine-tuned model $\epsilon_\theta$ over the reference model $\epsilon_{\text{ref}}$, specifically $r_\theta(\mathbf{c}, \mathbf{x}, t) \propto \delta_\theta(\mathbf{c}, \mathbf{x}_t, t)$, where

$$\delta_\theta(\mathbf{c}, \mathbf{x}_t, t) := -\left(\|\epsilon - \epsilon_\theta(\mathbf{x}_t, \mathbf{c}, t)\|_2^2 - \|\epsilon - \epsilon_{\text{ref}}(\mathbf{x}_t, \mathbf{c}, t)\|_2^2\right). \tag{2}$$

The resulting preference objective is

$$\mathcal{L}_{\text{Diff-DPO}}(\theta) = -\mathbb{E}_{(\mathbf{c}, \mathbf{x}^+, \mathbf{x}^-) \sim \mathcal{D}, t}\left[\log \sigma\left(\beta\,T\,\omega(\lambda_t)\left(\delta_\theta(\mathbf{c}, \mathbf{x}_t^+, t) - \delta_\theta(\mathbf{c}, \mathbf{x}_t^-, t)\right)\right)\right],$$

where $T$ is the number of diffusion steps and $\omega(\lambda_t)$ reweights timestep contributions.

## 4 METHOD

### 4.1 LISTWISE PREFERENCE OPTIMIZATION FOR DIFFUSION MODELS

Existing preference alignment methods for diffusion models, such as Diffusion-DPO, are built on the Bradley–Terry (BT) model, which assumes pairwise preferences. In this work, we adopt listwise preference optimization, $\mathbf{x}^{(1)} \succ \mathbf{x}^{(2)} \succ \cdots \succ \mathbf{x}^{(m)}$, which leverages the ranking as a more fine-grained information for preferences. To capture this higher-order structure, we adopt the *Plackett–Luce (PL)* model (Plackett, 1975), a probabilistic model for listwise rankings.

**Reward Under Plackett–Luce Model.** Let $\mathcal{G} = \{\mathbf{x}^{(1)}, \mathbf{x}^{(2)}, \ldots, \mathbf{x}^{(m)}\}$ denote a group of $m$ candidate images generated under prompt $\mathbf{c}$, ranked by a user as $\mathbf{x}^{(1)} \succ \mathbf{x}^{(2)} \succ \cdots \succ \mathbf{x}^{(m)}$. The PL model defines the probability of this ranking as

$$p_{\text{PL}}(\mathbf{x}^{(1)} \succ \mathbf{x}^{(2)} \succ \cdots \succ \mathbf{x}^{(m)} \mid \mathbf{c}) = \prod_{j=1}^{m} \frac{\exp\big(r(\mathbf{c}, \mathbf{x}^{(j)})\big)}{\sum_{k=j}^{m} \exp\big(r(\mathbf{c}, \mathbf{x}^{(k)})\big)},$$

where $r(\mathbf{c}, \mathbf{x})$ denotes the latent reward of the image $\mathbf{x}$ conditioned on the prompt $\mathbf{c}$. This formulation can be viewed as applying a softmax over each suffix sublist $\mathbf{x}^{(j)}, \mathbf{x}^{(j+1)}, \ldots, \mathbf{x}^{(m)}$, ensuring that at every step the selected item is assigned a higher preference than all remaining lower-ranked candidates.

**Objective for Diffusion-LPO** We would like to finetune the diffusion model $p_\theta(\mathbf{x}_0|\mathbf{c})$ with the RLHF objective as in Equation 1. For diffusion models, the KL-divergence regularizes the whole diffusion chain, and the reward is defined as $r(\mathbf{c}, \mathbf{x}_0) = \mathbb{E}_{p_\theta(\mathbf{x}_{1:T}|\mathbf{x}_0, \mathbf{c})}[R(\mathbf{c}, \mathbf{x}_{0:T})]$, where $(\mathbf{x}_1, ..., \mathbf{x}_T)$ are variables defined on the diffusion path. Maximizing the reward from the objective of Plackett-Luce model, we can get the listwise Diffusion-DPO objective:

$$\mathcal{L}_{\text{Diffusion-LPO}}(\theta) = - \mathbb{E}_{(\mathbf{c},\mathbf{x}_0^{(1:m)})\sim\mathcal{D},\, \mathbf{x}_{1:T}^{(1:m)}\sim p_\theta(\mathbf{x}_{1:T}^{(1:m)}|\mathbf{x}_0^{(1:m)})}$$
$$\sum_{j=1}^{m} \left[ \beta \, \log \frac{p_\theta(\mathbf{x}_{0:T}^{(j)} \mid \mathbf{c})}{p_{\text{ref}}(\mathbf{x}_{0:T}^{(j)} \mid \mathbf{c})} - \log \sum_{k=j}^{m} \exp\Big( \beta \, \log \frac{p_\theta(\mathbf{x}_{0:T}^{(k)} \mid \mathbf{c})}{p_{\text{ref}}(\mathbf{x}_{0:T}^{(k)} \mid \mathbf{c})}\Big) \right]. \tag{3}$$

Let $\epsilon_\theta(\cdot, \mathbf{c}, t)$ and $\epsilon_{\text{ref}}(\cdot, \mathbf{c}, t)$ be the learned and reference denoisers. Using the standard Diffusion-DPO surrogate $\Delta_\theta(\mathbf{x}_{0:T}^{(j)} \mid \mathbf{c}) \approx T\,\omega(\lambda_t)\,\delta_\theta(\mathbf{c}, \mathbf{x}_t^{(j)}, t)$, where $\delta_\theta(\mathbf{c}, \mathbf{x}_t, t)$ is defined in Equation 2, the listwise objective becomes

$$\mathcal{L}_{\text{Diffusion-LPO}}(\theta) = - \mathbb{E}_{(\mathbf{c},\mathbf{x}_0^{(1:m)})\sim\mathcal{D},\, \mathbf{x}_t^{(1:m)}\sim p_\theta(\mathbf{x}_t^{(1:m)}|\mathbf{x}_0^{(1:m)})}$$
$$\sum_{j=1}^{m} \left[ \beta\,T\,\omega(\lambda_t)\,\delta_\theta(\mathbf{c}, \mathbf{x}_t^{(j)}, t) - \log \sum_{k=j}^{m} \exp\Big( \beta\,T\,\omega(\lambda_t)\,\delta_\theta(\mathbf{c}, \mathbf{x}_t^{(k)}, t)\Big) \right]. \tag{4}$$

The proof of deriving Equation (4) from the RLHF objective (Equation 1) can be found in Appendix B.1. As listwise optimization is a more general form of pairwise, it is possible to extend Diffusion-LPO with other pairwise DPO methods. We use DSPO (Zhu et al., 2025) as an example and provide the derivation and relevant discussion in Appendix E. We observe improvement when using listwise preferences in DSPO compared with original pairwise DSPO, with detailed results shown in Appendix E.2.

Besides Diffusion-LPO, we also found similar existing works that also aim to optimize at the list level. We include a discussion of the comparison between our method and theirs in Appendix B.2.

**Constructing Listwise Groups.** The Pick-a-Pic dataset provides user preferences in a pairwise form, e.g., $\mathbf{x}_a \succ \mathbf{x}_b$ and $\mathbf{x}_b \succ \mathbf{x}_c$ under the same prompt $\mathbf{c}$. To construct a more faithful representation of human feedback, we form a list of preferences from the above pairwise preferences as $\mathbf{x}_a \succ \mathbf{x}_b \succ \mathbf{x}_c$. We conduct aggregation as directed acyclic graphs (DAG) of preferences and extract listwise paths, which we treat as ranking sublists. We provide statistics of transforming the pick-a-pic dataset into listwise in Appendix A. An example of the preference group can be found in Figure 5.

## 4.2 Advantage of Listwise Preference Modeling

Given a list of preferences, prior work on pairwise optimization (Wallace et al., 2024) can be extended naively. One can find the optimal policy model by minimizing the objective below, where we refer to it as Group Pairwise DPO(GP-DPO):

$$L_{\text{GP-DPO}} = -\mathbb{E}_{(\mathbf{c}, \mathbf{x}_0^{(1:m)}) \sim \mathcal{D}, \mathbf{x}_{1:T}^{(1:m)} \sim p_\theta(\mathbf{x}_{1:T}^{(1:m)} | \mathbf{x}_0^{(1:m)}, \mathbf{c})} \sum_{1 \le j < k \le m} \log \sigma\left(r(\mathbf{c}, \mathbf{x}^{(j)}) - r(\mathbf{c}, \mathbf{x}^{(k)})\right). \quad (5)$$

Such an objective is also derived in Chen et al. (2025). Conducting pairwise optimization in a group of size $m$ will introduce $m(m-1)/2$ pairs for comparison with equal importance. For our proposed method, denote the score $s_\theta^{(j)} := \frac{p_\theta(\mathbf{x}_{0:T}^{(j)} | \mathbf{c})}{p_{\text{ref}}(\mathbf{x}_{0:T}^{(j)} | \mathbf{c})}$, which is the part of reward $R(\mathbf{c}, \mathbf{x}_{0:T}^{(j)})$ related to policy model up to scalar $\beta$. In objective (4), when treating $\mathbf{x}^{(j)}$ as the positive sample and optimizing over its set of negative samples $\{\mathbf{x}^{(j+1)}, \cdots, \mathbf{x}^{(m)}\}$, it yields the following reward difference

$$P_{\text{Diffusion-LPO}}(\mathbf{x}^{(j)} > \mathbf{x}^{(k)}, \forall k \in \{j+1, \cdots, m\}) \propto \beta \log s_\theta^{(j)} - \log(\sum_{k=j}^{m} (s_\theta^{(k)})^\beta). \quad (6)$$

Besides, we can find for Group Pairwise DPO with objective (5), the reward difference when treating $\mathbf{x}^{(j)}$ as the positive sample and computing with all the preference pairs as

$$P_{\text{GP-DPO}}(\mathbf{x}^{(j)} > \mathbf{x}^{(k)}, \forall k \in \{j+1, \cdots, m\}) \propto \beta \log s_\theta^{(j)} - \frac{1}{m-j+1} \sum_{k=j+1}^{m} \log((s^{(j)})^\beta + (s^{(k)})^\beta). \quad (7)$$

The detailed analysis can be found in Appendix B.3. GP-DPO and Diffusion-LPO derive different reward functions in terms of human preference of $\mathbf{x}^{(j)}$ over the negative samples. Compare their rewards for negative samples, we can find

$$\frac{1}{m-j+1} \sum_{k=j+1}^{m} \log((s^{(j)})^\beta + (s^{(k)})^\beta) \le \frac{1}{m-j+1} \sum_{k=j+1}^{m} \max_{k:k>j} \log((s^{(j)})^\beta + (s^{(k)})^\beta)$$
$$\le \log\left(\sum_{k=j}^{m} (s_\theta^{(k)})^\beta\right). \quad (8)$$

From the above analysis, we observe that GP-DPO leads to an underestimate of the aggregate reward of negative samples, since it reduces the ranking into equally weighted pairwise comparisons as shown in (7) . In particular, the reward assigned to negatives under GP-DPO is upper-bounded by the corresponding term in Diffusion-LPO, as shown in (8). This underestimation inflates the margin between the current positive sample $\mathbf{x}^{(j)}$ and its negatives. In contrast, Diffusion-LPO directly normalizes over the entire negative group as shown in (6) , which ensures higher-ranked samples are favored over the lower-ranked ones.

## 5 Experiment

### 5.1 Experiment Setup

**Models, Datasets, and Evaluations.** We finetune two text-to-image diffusion backbones: Stable Diffusion 1.5 (SD1.5) (Rombach et al., 2022) and SDXL (Podell et al., 2023). Training data is drawn from the Pick-a-Pic v1 preference dataset (Kirstain et al., 2023), and our listwise data construction follows the procedure described in Appendix A. In our Diffusion-LPO implementation, the extracted preference lists have variable lengths, and we cap the maximum list length at 8, so Diffusion-LPO is trained on lists of size from 2 to 8; when the list size is 2, the objective reduces to the standard pairwise Diffusion-DPO loss. We evaluate models on: (1) **General Text-to-Image Alignment**: We use the prompts from three datasets for evaluation: Pick-a-Pic test dataset (Kirstain et al., 2023), Parti-Prompts (Yu et al., 2022), HPSV2 test dataset (Wu et al., 2023). (2) **Image Editing**: We use two image editing datasets, InstructPix2Pix (Brooks et al., 2023) and ImgEdit benchmark (Ye et al., 2025) for evaluation. (3) **Personalized Preference Generation**: Testing whether models adapt to

individual user preference embeddings. We train the SD1.5 model with the pipeline in PPD (Dang et al., 2025) and evaluate the personalization performance on the Pick-a-Pic test dataset. Comprehensive details of the evaluation pipeline and prompts are given in Appendix C.1. Implementation details are in Appendix C.2. Computation overhead discussion is in Appendix C.4.

**Baselines.** We consider baselines of: SFT (supervised fine-tuning on preferred images), pairwise Diffusion-DPO (Wallace et al., 2024), and DSPO (Zhu et al., 2025), along with the original pretrained SD1.5 and SDXL. Besides finetuning on these baselines, we also include Diffusion-DPO*, SD1.5, and SDXL tuned on the Pick-a-Pic v2 preference dataset, which has almost twice the data size as Pick-a-Pic v1. We directly use the released checkpoint for evaluation. Notice that we only consider baselines that do not require additional reward models or evaluators for a fair comparison.

**Evaluation.** For text-to-image alignment, we adopt five widely used automatic evaluators: PickScore (Kirstain et al., 2023), HPSV2 (Wu et al., 2023), CLIP score (Radford et al., 2021), Image Reward (Xu et al., 2023), and Aesthetic Score (AES) (Schuhmann, 2022). For image-editing task, we adopt three metrics for the InstructPix2Pix dataset: L1, CLIP, and DINO. L1 calculates the pixel-level absolute difference between the generated image and ground truth target image. CLIP and DINO reflect the edited image quality by measuring the cosine similarity of the CLIP (Radford et al., 2021) and DINO (Caron et al., 2021) embeddings between the generated images and ground truth target images. We use the GPT4o (Hurst et al., 2024) to evaluate the images generated under ImgEdit benchmark, following its suggested evaluation pipeline. For personalized preference generation, where no standard automatic metric exists, we follow recent evaluation practice in (Dang et al., 2025) and rely on GPT-4o (Hurst et al., 2024) as a judge, prompting it to rate which generated image better aligns with a user's historical preferences.

## 5.2 GENERAL EVALUATION: TEXT-TO-IMAGE ALIGNMENT

Table 1: Winrate results over original SD1.5 and SDXL for Diffusion-LPO compared with other baselines. For abbreviation, we use "PS", "IM", and "AES" to represent PickScore, Image Reward, and Aesthetics score. "Diff." is short for "Diffusion". **Note: best results trained under Pick-a-Pic v1 (Kirstain et al., 2023) are in boldface. Diffusion-DPO* is trained under Pick-a-Pic v2 dataset, with almost twice the data size than v1**. We directly use the checkpoint trained by (Wallace et al., 2024). All results are averaged over generations from 5 random seeds. ↑: higher values indicate better performance.

| Dataset | Method | Stable Diffusion 1.5 | | | | | Stable Diffusion XL | | | | |
|---|---|---|---|---|---|---|---|---|---|---|---|
| | | PS ↑ | HPS ↑ | CLIP↑ | IM↑ | AES↑ | PS↑ | HPS ↑ | CLIP↑ | IM ↑ | AES ↑ |
| Pick-a-Pic | SFT | 73.4% | 80.7% | 56.3% | 74.9% | **71.7%** | 22.5% | 39.3% | 46.6% | 37.0% | 24.3% |
| | Diff.-DPO* | 73.3% | 69.8% | 57.1% | 61.7% | 63.4% | 77.1% | 78.4% | 66.0% | 70.0% | 50.8% |
| | Diff.-DPO | 68.2% | 70.7% | 54.8% | 65.2% | 60.1% | 73.0% | 81.1% | **64.5%** | 66.4% | 47.1% |
| | DSPO | 73.1% | **82.6%** | 57.8% | **75.2%** | 70.9% | 59.2% | 77.4% | 56.6% | 62.9% | 45.9% |
| | Diff.-LPO | **80.4%** | 74.6% | **58.5%** | 68.1% | 64.0% | **77.1%** | **87.1%** | 64.0% | **73.7%** | **51.5%** |
| Parti-Prompts | SFT | 65.0% | 79.0% | 54.2% | 69.7% | **72.3%** | 24.2% | 39.6% | 47.9% | 40.9% | 32.0% |
| | Diff.-DPO* | 67.3% | 64.9% | 53.7% | 60.2% | 62.3% | 70.0% | 80.5% | 64.6% | 74.0% | 58.6% |
| | Diff.-DPO | 59.7% | 64.0% | 52.9% | 61.6% | 58.9% | 66.8% | 77.7% | 56.0% | 68.7% | 56.0% |
| | DSPO | 63.8% | 78.2% | 54.3% | **70.3%** | 71.1% | 55.5% | 78.1% | 54.2% | 65.5% | 55.4% |
| | Diff.-LPO | **71.9%** | 69.7% | **57.4%** | 65.9% | 62.5% | **72.8%** | **82.9%** | **60.1%** | **73.6%** | **59.7%** |
| HPSV2 | SFT | 75.1% | 85.0% | 56.4% | **77.7%** | **72.9%** | 20.4% | 40.1% | 47.8% | 42.3% | 25.9% |
| | Diff.-DPO* | 76.5% | 71.0% | 57.4% | 64.4% | 67.2% | 72.5% | 79.2% | 59.3% | 70.4% | 55.5% |
| | Diff.-DPO | 69.5% | 70.9% | 51.5% | 65.8% | 62.0% | 72.1% | 80.0% | **59.2%** | 70.8% | 47.7% |
| | DSPO | 74.4% | **85.3%** | 56.8% | 77.4% | 71.6% | 58.4% | 77.3% | 47.8% | 63.7% | 48.2% |
| | Diff.-LPO | **82.9%** | 76.6% | **57.7%** | 69.0% | 65.1% | **74.6%** | **85.0%** | 56.3% | **72.3%** | **51.9%** |

We report text-to-image generation results in Table 1. Diffusion-LPO achieves substantial gains over its pairwise baselines and even surpasses Diffusion-DPO trained on Pick-a-Pic v2, on most metrics. On SD1.5, Diffusion-LPO shows clear improvements over pairwise Diffusion-DPO. In particular, the PickScore increases by more than 12% across all evaluation sets, indicating more reliable alignments with human preferences. On SDXL, Diffusion-LPO consistently outperforms all

baselines. Notably, Diffusion-LPO improves the PickScore win rate on Parti-prompts with 6% and HPS score win rate on HPSV2 with 5% compared with its pairwise baseline, Diffusion-DPO. While SFT achieves competitive results on SD1.5, its performance drops significantly on SDXL, with win rates falling below 50%. This degradation is likely due to a mismatch between the relatively lower quality of training data and the higher intrinsic quality of SDXL generations. In contrast, Diffusion-LPO exploits relative ranking information effectively, enabling the model to better capture human preferences and outperform its pairwise baseline, Diffusion-DPO.

Figure 3 qualitatively illustrates these improvements. Diffusion-LPO demonstrates a stronger capability to handle fine-grained structures. For the last column, although all methods produce visually appealing images of the dolphin mid-jump, Diffusion-LPO generates dolphin tails with smoother curvature and more accurate shape, highlighting its superiority in capturing detailed features. More qualitative illustrations of Diffusion-LPO over other baselines are shown in Appendix D.1. We also report additional results of training DiT family diffusion models in Appendix F.1. We fine-tune the SD3.5-Medium (Esser et al., 2024) model on Diffusion-DPO and Diffsion-LPO. Diffusion-LPO consistently outperforms Diffusion-DPO, demonstrating the effectiveness of our method on both traditional U-Net backbones and modern DiT-based diffusion architectures.

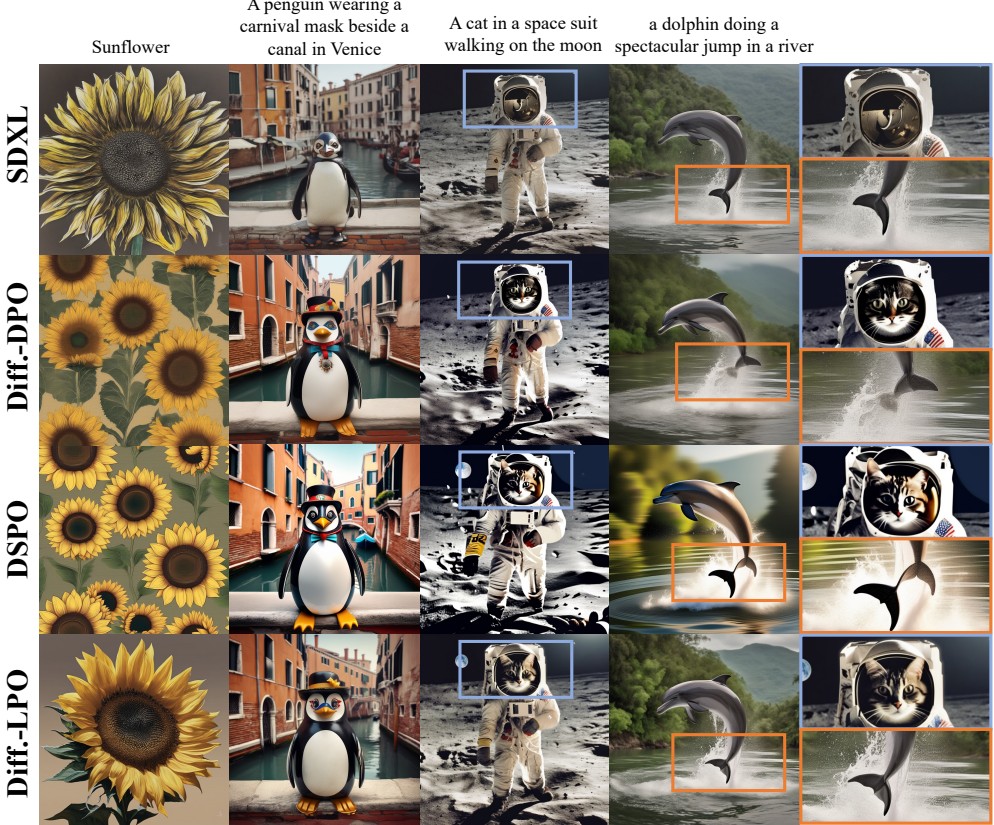

Figure 3: Images generated from original SDXL, Diffusion-DPO, DSPO, and Diffusion-LPO. Diffusion-LPO demonstrates improved image generation quality over other baselines regarding general aesthetics and detail handling. The last column indicates the zoom-in parts.

## 5.3 IMAGE EDITING

We further evaluate Diffusion-LPO in the context of instruction-based image editing. Following prior work, we adopt the InstructPix2Pix dataset (Brooks et al., 2023), which provides paired source images, natural language edit instructions, and target outputs. We perform editing with Stable Diffusion 1.5, comparing the original model and its tuned variants. For editing, we apply SDEdit (Meng et al., 2021) with a noise strength of 0.6. Diffusion-LPO achieves notable improvements over its pairwise counterpart, Diffusion-DPO, with win-rate gains of 4.3% on DINO score, 3.6% on CLIP

score, and 3.8% on L1 distance respectively. We also adopt a more recent image editing benchmark Ye et al. (2025), which includes both single-turn and multi-turn image editing instructions, and use GPT4o to evaluate the image editing result. Diffusion-LPO achieves winrate of 56.3% compared to images generated from Diffusion-DPO. These results indicate that listwise supervision provides stronger alignment signals for following editing instructions. Qualitative results are shown in Figure 8 in Appendix D.2.

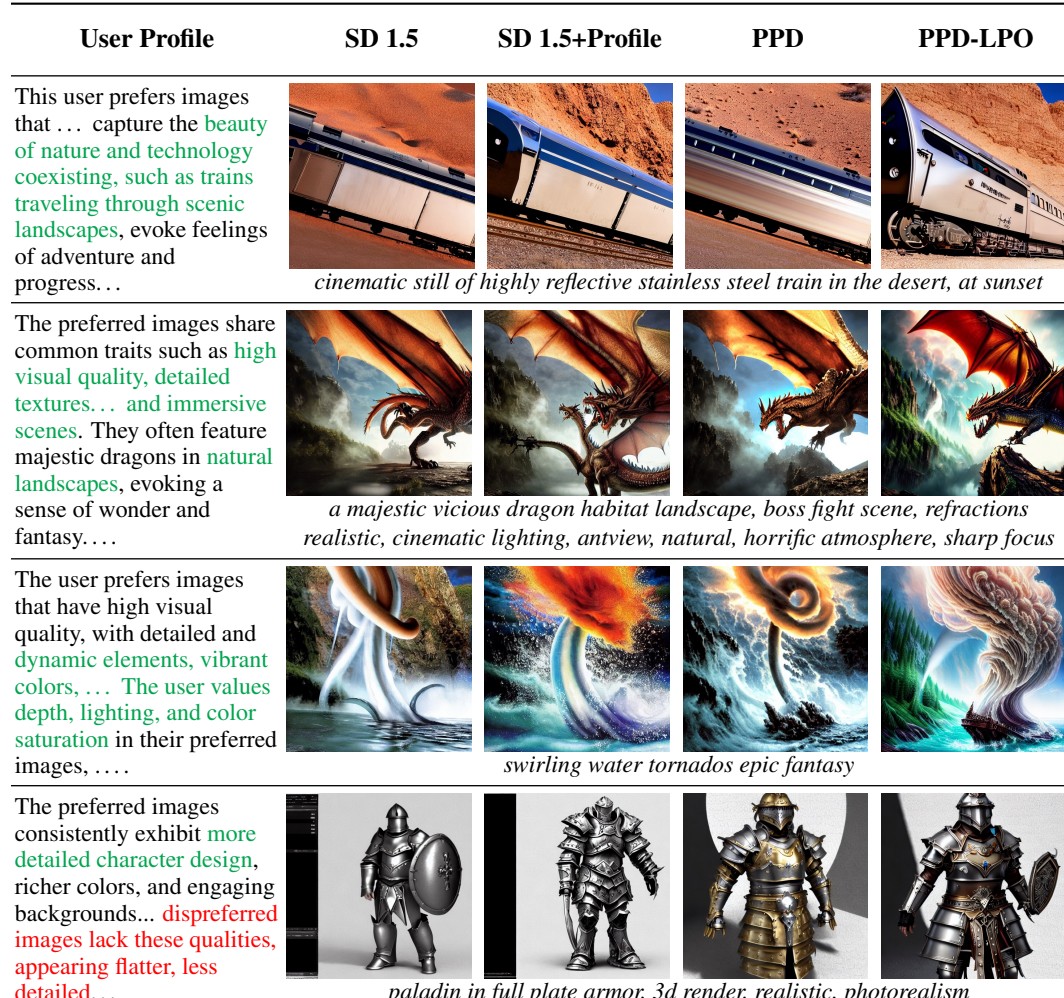

| User Profile | SD 1.5 | SD 1.5+Profile | PPD | PPD-LPO |
|---|---|---|---|---|

This user prefers images that ... capture the beauty of nature and technology coexisting, such as trains traveling through scenic landscapes, evoke feelings of adventure and progress...

*cinematic still of highly reflective stainless steel train in the desert, at sunset*

The preferred images share common traits such as high visual quality, detailed textures... and immersive scenes. They often feature majestic dragons in natural landscapes, evoking a sense of wonder and fantasy....

*a majestic vicious dragon habitat landscape, boss fight scene, refractions realistic, cinematic lighting, antview, natural, horrific atmosphere, sharp focus*

The user prefers images that have high visual quality, with detailed and dynamic elements, vibrant colors, ... The user values depth, lighting, and color saturation in their preferred images, ....

*swirling water tornados epic fantasy*

The preferred images consistently exhibit more detailed character design, richer colors, and engaging backgrounds... dispreferred images lack these qualities, appearing flatter, less detailed...

*paladin in full plate armor, 3d render, realistic, photorealism*

Figure 4: Images generated under Diffusion-LPO with personal preference alignment with other baselines. User profiles are summarized by VLM. "SD 1.5+Profile" represents images generated using SD 1.5 with user profile appended to the caption. We highlight the user preferences in green and dispreferences in red.

## 5.4 PERSONALIZED PREFERENCE ALIGNMENT

Beyond general text-to-image alignment, we further evaluate Diffusion-LPO under the personalized preference setting. Following the pipeline of Dang et al. (2025), we condition the diffusion model on user embeddings extracted by a vision–language model that summarizes each user's historical preferences (see Appendix C.3 for details).

We use SD1.5 as the backbone and evaluate it on the Pick-a-Pic test set, which is partitioned into held-in users (seen during training) and held-out users (unseen during training). As a baseline, we adopt Personalized Preference Diffusion (PPD) (Dang et al., 2025), which aligns user embeddings with diffusion models using the Diffusion-DPO objective. We then replace the pairwise loss with

our listwise formulation, denoted as PPD-LPO. Incorporating Diffusion-LPO into the personalization pipeline yields consistent gains: the win rate improves from 71.1% to 72.3% on held-in users and from 70.3% to 80.2% on held-out users, demonstrating that listwise optimization both enhances personalization for known users and substantially improves generalization to unseen users. Qualitative analysis is shown in Figure 4. While the original PPD improves alignment relative to the untuned SD1.5 model, PPD-LPO further enhances both visual appeal and fine-grained detail.

## 6 ABLATION

### 6.1 RANK ENFORCEMENT FROM DIFFUSION-LPO

To further assess the effectiveness of Diffusion-LPO, we compare it against Group-Pairwise DPO (GP-DPO), with the objective in Equation 5 to finetune Stable Diffusion 1.5. Both objectives are trained under the same objective. For each prompt, we calculate the scores of images generated by each evaluator. Table 2 reports, for each evaluator, the win rate of Diffusion-LPO over GP-DPO, where the win rate is defined as the percentage of prompts on which images from Diffusion-LPO receive higher scores than those from GP-DPO. Since Diffusion-LPO outperforms GP-DPO on most metrics and evaluators, this indicates that explicitly optimizing the model on listwise human preference data leads to better alignment quality, consistent with our theoretical analysis in Section 4.2. We provide the corresponding significance tests in Appendix F.2. For all metrics except CLIP, the improvement of Diffusion-LPO over GP-DPO is statistically significant.

Table 2: Winrate for Diffusion-LPO on SD1.5 in comparison to SD1.5 trained under GP-DPO.

| Dataset | PS↑ | HPS ↑ | CLIP ↑ | IM ↑ | AES ↑ |
|---------|-----|-------|--------|------|-------|
| Pick-a-Pic | $52.3\%_{\pm 0.012}$ | $51.3\%_{\pm 0.010}$ | $50.6\%_{\pm 0.014}$ | $50.8\%_{\pm 0.027}$ | $53.7\%_{\pm 0.011}$ |
| HPSv2 | $52.4\%_{\pm 0.012}$ | $54.0\%_{\pm 0.015}$ | $50.6\%_{\pm 0.018}$ | $51.3\%_{\pm 0.008}$ | $49.1\%_{\pm 0.020}$ |
| Parti-Prompts | $52.4\%_{\pm 0.012}$ | $53.0\%_{\pm 0.017}$ | $50.8\%_{\pm 0.012}$ | $51.4\%_{\pm 0.012}$ | $52.1\%_{\pm 0.010}$ |

### 6.2 DIFFERENT MAXIMUM LIST SIZE

To assess the influence of the maximum list length, we compare performance under maximum list sizes of 4, 8, and 12 in Table 3. The result shows that, increasing maximum list size from 4 to 8 brings 1% of overall improvement, while increasing list size from 8 to 12 yields only marginal differences. This suggests that list size 8 for Diffusion-LPO in our setting is sufficiently large, which is also consistent with the empirical distribution of our data, where approximately 95% of preference lists have length at most 8.

Table 3: Winrate for Diffusion-LPO on SD1.5 vs SD1.5, under different maximum list size. Results reported on Pick-a-Pic test dataset.

| Max List Size | Avg↑ | PS↑ | HPS ↑ | CLIP ↑ | IM ↑ | AES ↑ |
|---------------|------|-----|-------|--------|------|-------|
| $m = 4$ | 68.2% | 79.4% | 73.0% | 58.5% | 68.5% | 61.6% |
| $m = 8$ | 69.1% | 80.4% | 74.6% | 58.5% | 68.1% | 64.0% |
| $m = 12$ | 69.3% | 79.9% | 74.7% | 60.5% | 69.2% | 62.1% |

## 7 CONCLUSION

In this work, we introduce Diffusion-LPO, a novel framework for aligning text-to-image models using listwise preference optimization. By modeling full preference rankings with the Plackett-Luce model, our approach surpasses pairwise methods like Direct Preference Optimization (DPO). Experiments on Stable Diffusion 1.5 and SDXL demonstrate that Diffusion-LPO produces outputs with superior coherence and human preference alignment, highlighting the efficacy of listwise supervision for fine-tuning generative models.

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

## A  PICK-A-PIC DATASET LISTWISE CONSTRUCTION

We restructure the Pick-a-Pic dataset into user–prompt specific directed acyclic graphs (DAGs) of image rankings. Specifically, we collect all images generated under the same prompt and annotated by the same user, and construct a graph where each image corresponds to a node. Whenever the user provides a pairwise annotation indicating that image $\mathbf{x}_A$ is preferred over image $\mathbf{x}_B$, we add a directed edge $\mathbf{x}_A \rightarrow \mathbf{x}_B$. By aggregating all such edges, we obtain a DAG that encodes the transitive structure of the user's annotations. From these DAGs, we can extract consistent listwise rankings of images (e.g. $\mathbf{x}_A \succ \mathbf{x}_B \succ \mathbf{x}_C$) whenever sufficient pairwise information is available, thus transforming independent pairwise judgments into richer listwise preference orders.

Table 4: Statistics of Pick-a-Pic after listwise group construction.

| List Size | # of list |
|-----------|-----------|
| 2 | 225,656 |
| 3 | 81,704 |
| 4 | 42,113 |
| 5 | 22,216 |
| 6 | 14,191 |
| 7 | 8,953 |
| $\geq 8$ | 23,087 |

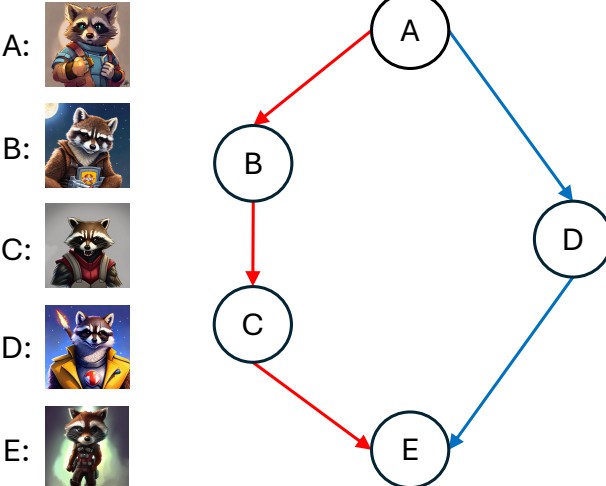

Figure 5: An example of a group with preferences that forms a DAG. The arrow pointing from image $\mathbf{x}_A$ to image $\mathbf{x}_B$ represents human preference: $\mathbf{x}_A \succ \mathbf{x}_B$. The prompt is "Rocket Raccoon, furry art, fanart, digital painting". Here, valid rank list will be $(\mathbf{x}_A \succ \mathbf{x}_B \succ \mathbf{x}_C \succ \mathbf{x}_E)$ and $(\mathbf{x}_A \succ \mathbf{x}_D \succ \mathbf{x}_E)$.

**Data usability.** Table 4 summarizes the statistics of the reformulated lists. In total, we analyze 511,840 preference pairs. Among them, 225,656 pairs (44.09%) form lists of size 2, which naturally reduce to standard pairwise DPO training. Importantly, 274,895 pairs (53.71%) belong to lists of size > 2, making them directly applicable to listwise preference optimization. Only 11,289 pairs (2.20%) fall into inconsistent groups and are discarded. Such inconsistencies correspond to cyclic preferences (e.g., $\mathbf{x}_A \succ \mathbf{x}_B$, $\mathbf{x}_B \succ \mathbf{x}_C$, $\mathbf{x}_C \succ \mathbf{x}_A$), which cannot be represented as DAGs and thus reflect low-quality or noisy annotations. The very small fraction of such cases demonstrates that the overwhelming majority of human annotations are internally consistent and can be reliably aggregated into rankings. In our training, lists of size 2 still contribute through Diffusion-DPO, while longer lists provide richer preference information for Diffusion-LPO.

**Constructing lists from DAGs.** Each group is represented as a directed acyclic graph (DAG), where nodes correspond to images and edges encode user-indicated pairwise preferences. To transform this structure into training examples for listwise optimization, we enumerate valid topological paths within each DAG. Each path yields an ordered list of images $\{x^{(1)} \succ x^{(2)} \succ \cdots \succ x^{(m)}\}$ that is consistent with all local pairwise edges. These lists are then used directly in the Plackett–Luce likelihood of our objective. In practice, multiple valid paths may exist within a single DAG; we use all valid paths for Diffusion-LPO.

# B DIFFUSION-LPO OBJECTIVE

## B.1 DIFFUSION-LPO OBJECTIVE

Recall the RLHF objective:

$$\max_{p_\theta} \mathbb{E}_{\mathbf{c},\mathbf{x}_0} \left[ r(\mathbf{c}, \mathbf{x}_0) \right] - \beta \mathbb{D}_{\mathrm{KL}} \left[ p_\theta(\mathbf{x}_0|\mathbf{c}) \| p_{\mathrm{ref}}(\mathbf{x}_0|\mathbf{c}) \right], \tag{9}$$

where $\beta$ is a parameter that controls how much $p_\theta(\mathbf{x}_0|\mathbf{c})$ deviates from $p_{\mathrm{ref}}(\mathbf{x}_0|\mathbf{c})$.

We define $R(\mathbf{c}, \mathbf{x}_{0:T})$ as the reward on the whole diffusion chain to define the reward $r(\mathbf{c}, \mathbf{x}_0)$ as $r(\mathbf{c}, \mathbf{x}_0) = \mathbb{E}_{p(\mathbf{x}_{1:T}|\mathbf{x}_0,\mathbf{c})}[R(\mathbf{c}, \mathbf{x}_{0:T})]$. Given 9, we have

$$
\begin{aligned}
&\min_{p_\theta} \; -\mathbb{E}_{p_\theta(\mathbf{x}_0|\mathbf{c})} \left[ \tfrac{1}{\beta} r(\mathbf{c}, \mathbf{x}_0) \right] + \mathbb{D}_{\mathrm{KL}} \left[ p_\theta(\mathbf{x}_0|\mathbf{c}) | p_{\mathrm{ref}}(\mathbf{x}_0|\mathbf{c}) \right] \\
\leq \;&\min_{p_\theta} \; -\mathbb{E}_{p_\theta(\mathbf{x}_{0:T}|\mathbf{c})} \left[ \tfrac{1}{\beta} R(\mathbf{c}, \mathbf{x}_{0:T}) \right] + \mathbb{D}_{\mathrm{KL}} \left[ p_\theta(\mathbf{x}_{0:T}|\mathbf{c}) | p_{\mathrm{ref}}(\mathbf{x}_{0:T}|\mathbf{c}) \right] \\
= \;&\min_{p_\theta} \; \mathbb{E}_{p_\theta(\mathbf{x}_{0:T}|\mathbf{c})} \left[ \log \frac{p_\theta(\mathbf{x}_{0:T}|\mathbf{c})}{p_{\mathrm{ref}}(\mathbf{x}_{0:T}|\mathbf{c})} - \tfrac{1}{\beta} R(\mathbf{c}, \mathbf{x}_{0:T}) \right] \\
= \;&\min_{p_\theta} \; \mathbb{E}_{p_\theta(\mathbf{x}_{0:T}|\mathbf{c})} \left[ \log \frac{p_\theta(\mathbf{x}_{0:T}|\mathbf{c})}{p_{\mathrm{ref}}(\mathbf{x}_{0:T}|\mathbf{c}) \exp\left( R(\mathbf{c}, \mathbf{x}_{0:T})/\beta \right)} \right] \\
= \;&\min_{p_\theta} \; \mathbb{E}_{p_\theta(\mathbf{x}_{0:T}|\mathbf{c})} \left[ \log \frac{p_\theta(\mathbf{x}_{0:T}|\mathbf{c})}{\frac{1}{Z(\mathbf{c})} p_{\mathrm{ref}}(\mathbf{x}_{0:T}|\mathbf{c}) \exp\left( R(\mathbf{c}, \mathbf{x}_{0:T})/\beta \right)} - \log Z(\mathbf{c}) \right] \\
= \;&\min_{p_\theta} \; \mathbb{D}_{\mathrm{KL}} \left[ p_\theta(\mathbf{x}_{0:T}|\mathbf{c}) \| p_{\mathrm{ref}}(\mathbf{x}_{0:T}|\mathbf{c}) \exp(R(\mathbf{c}, \mathbf{x}_{0:T})/\beta)/Z(\mathbf{c}) \right].
\end{aligned}
\tag{10}
$$

where $Z(\mathbf{c}) = \sum_{\mathbf{x}} p_{\mathrm{ref}}(\mathbf{x}_{0:T}|\mathbf{c}) \exp\left( r(\mathbf{c}, \mathbf{x}_0)/\beta \right)$ is the partition function, and we get rid of $\log Z(\mathbf{c})$ as it is independent from $p_\theta$. The optimal $p_\theta^*(\mathbf{x}_{0:T}|\mathbf{c})$ of Equation (10) has a unique closed-form solution:

$$p_\theta^*(\mathbf{x}_{0:T}|\mathbf{c}) = p_{\mathrm{ref}}(\mathbf{x}_{0:T}|\mathbf{c}) \exp(R(\mathbf{c}, \mathbf{x}_{0:T})/\beta)/Z(\mathbf{c}),$$

Therefore, we have the reparameterization of the reward function

$$R(\mathbf{c}, \mathbf{x}_{0:T}) = \beta \log \frac{p_\theta^*(\mathbf{x}_{0:T}|\mathbf{c})}{p_{\mathrm{ref}}(\mathbf{x}_{0:T}|\mathbf{c})} + \beta \log Z(\mathbf{c}).$$

Plugging this into the definition of $r$, we have the following

$$r(\mathbf{c}, \mathbf{x}_0) = \beta \mathbb{E}_{p_\theta(\mathbf{x}_{1:T}|\mathbf{x}_0,\mathbf{c})} \left[ \log \frac{p_\theta^*(\mathbf{x}_{0:T}|\mathbf{c})}{p_{\mathrm{ref}}(\mathbf{x}_{0:T}|\mathbf{c})} \right] + \beta \log Z(\mathbf{c}).$$

Substituting this reward reparameterization into the maximum likelihood objective of the Plackett-Luce model, the partition function cancels out, and we get a maximum likelihood objective defined on diffusion models, for the group $(\mathbf{c}, \mathbf{x}_0^{(1)}, ..., \mathbf{x}_0^{(m)})$:

$$\mathcal{L}_{\mathrm{Diffusion\text{-}LPO}}(\theta) = \log \prod_{j=1}^{m} \frac{\exp\left( \beta \mathbb{E}_{p_\theta(\mathbf{x}_{1:T}^{(j)}|\mathbf{x}_0^{(j)},\mathbf{c})} \left[ \log \frac{p_\theta^*(\mathbf{x}_{0:T}^{(j)}|\mathbf{c})}{p_{\mathrm{ref}}(\mathbf{x}_{0:T}^{(j)}|\mathbf{c})} \right] \right)}{\sum_{k=j}^{m} \exp\left( \beta \mathbb{E}_{p_\theta(\mathbf{x}_{1:T}^{(k)}|\mathbf{x}_0^{(k)},\mathbf{c})} \left[ \log \frac{p_\theta^*(\mathbf{x}_{0:T}^{(k)}|\mathbf{c})}{p_{\mathrm{ref}}(\mathbf{x}_{0:T}^{(k)}|\mathbf{c})} \right] \right)}.$$

By applying Jensen's inequality, we can push the expectation out, and with some simplification, we can have,

$$\mathcal{L}_{\text{Diffusion-LPO}}(\theta) = - \mathbb{E}_{(\mathbf{c}, \mathbf{x}_0^{(1:m)}) \sim \mathcal{D}, \, \mathbf{x}_{1:T}^{(1:m)} \sim p_\theta(\mathbf{x}_{1:T}^{(1:m)} | \mathbf{x}_0^{(1:m)})}$$
$$\sum_{j=1}^{m} \left[ \beta \, \log \frac{p_\theta(\mathbf{x}_{0:T}^{(j)} \mid \mathbf{c})}{p_{\text{ref}}(\mathbf{x}_{0:T}^{(j)} \mid \mathbf{c})} - \log \sum_{k=j}^{m} \exp\left( \beta \, \log \frac{p_\theta(\mathbf{x}_{0:T}^{(k)} \mid \mathbf{c})}{p_{\text{ref}}(\mathbf{x}_{0:T}^{(k)} \mid \mathbf{c})} \right) \right].$$

Since sampling from $p_\theta(\mathbf{x}_{1:T}|\mathbf{x}_0, \mathbf{c})$ is intractable, we utilize the forward process $q(\mathbf{x}_{1:T}|\mathbf{x}_0)$ for approximation. As $\mathbf{x}_{1:T}^{(j)} \sim q(\mathbf{x}_{1:T} \mid \mathbf{x}_0^{(j)}, \mathbf{c})$ for $j = 1, \ldots, m$,

$$L_{\text{approx}}(\theta) = -\sum_{j=1}^{m} \left[ \beta \, \mathbb{E}_{\mathbf{x}_{1:T}^{(j)}} \left[ \log \frac{p_\theta(\mathbf{x}_{0:T}^{(j)}|\mathbf{c})}{p_{\text{ref}}(\mathbf{x}_{0:T}^{(j)}|\mathbf{c})} \right] - \log \sum_{k=j}^{m} \exp\left( \beta \, \mathbb{E}_{\mathbf{x}_{1:T}^{(k)}} \left[ \log \frac{p_\theta(\mathbf{x}_{0:T}^{(k)}|\mathbf{c})}{p_{\text{ref}}(\mathbf{x}_{0:T}^{(k)}|\mathbf{c})} \right] \right) \right]$$

$$= -\sum_{j=1}^{m} \left[ \beta \, \mathbb{E}_{\mathbf{x}_{1:T}^{(j)}} \left[ \sum_{t=1}^{T} \log \frac{p_\theta(\mathbf{x}_{t-1}^{(j)} \mid \mathbf{x}_t^{(j)}, \mathbf{c})}{p_{\text{ref}}(\mathbf{x}_{t-1}^{(j)} \mid \mathbf{x}_t^{(j)}, \mathbf{c})} \right] \right.$$
$$\left. - \log \sum_{k=j}^{m} \exp\left( \beta \, \mathbb{E}_{\mathbf{x}_{1:T}^{(k)}} \left[ \sum_{t=1}^{T} \log \frac{p_\theta(\mathbf{x}_{t-1}^{(k)} \mid \mathbf{x}_t^{(k)}, \mathbf{c})}{p_{\text{ref}}(\mathbf{x}_{t-1}^{(k)} \mid \mathbf{x}_t^{(k)}, \mathbf{c})} \right] \right) \right]$$

$$= -\sum_{j=1}^{m} \left[ \beta \, \mathbb{E}_{\mathbf{x}_{1:T}^{(j)}} \left[ T \, \mathbb{E}_t \, \log \frac{p_\theta(\mathbf{x}_{t-1}^{(j)} \mid \mathbf{x}_t^{(j)}, \mathbf{c})}{p_{\text{ref}}(\mathbf{x}_{t-1}^{(j)} \mid \mathbf{x}_t^{(j)}, \mathbf{c})} \right] \right.$$
$$\left. - \log \sum_{k=j}^{m} \exp\left( \beta \, \mathbb{E}_{\mathbf{x}_{1:T}^{(k)}} \left[ T \, \mathbb{E}_t \, \log \frac{p_\theta(\mathbf{x}_{t-1}^{(k)} \mid \mathbf{x}_t^{(k)}, \mathbf{c})}{p_{\text{ref}}(\mathbf{x}_{t-1}^{(k)} \mid \mathbf{x}_t^{(k)}, \mathbf{c})} \right] \right) \right]$$

$$= -\sum_{j=1}^{m} \left[ \beta T \, \mathbb{E}_{t, \, \mathbf{x}_{t-1,t}^{(j)}} \left[ \log \frac{p_\theta(\mathbf{x}_{t-1}^{(j)} \mid \mathbf{x}_t^{(j)}, \mathbf{c})}{p_{\text{ref}}(\mathbf{x}_{t-1}^{(j)} \mid \mathbf{x}_t^{(j)}, \mathbf{c})} \right] \right.$$
$$\left. - \log \sum_{k=j}^{m} \exp\left( \beta T \, \mathbb{E}_{t, \, \mathbf{x}_{t-1,t}^{(k)}} \left[ \log \frac{p_\theta(\mathbf{x}_{t-1}^{(k)} \mid \mathbf{x}_t^{(k)}, \mathbf{c})}{p_{\text{ref}}(\mathbf{x}_{t-1}^{(k)} \mid \mathbf{x}_t^{(k)}, \mathbf{c})} \right] \right) \right]$$

$$= -\sum_{j=1}^{m} \left[ \beta T \, \mathbb{E}_{t, \, \mathbf{x}_t^{(1:m)}, \mathbf{x}_{t-1}^{(j)}} \left[ \log \frac{p_\theta(\mathbf{x}_{t-1}^{(j)} \mid \mathbf{x}_t^{(j)}, \mathbf{c})}{p_{\text{ref}}(\mathbf{x}_{t-1}^{(j)} \mid \mathbf{x}_t^{(j)}, \mathbf{c})} \right] \right.$$
$$\left. - \log \sum_{k=j}^{m} \exp\left( \beta T \, \mathbb{E}_{t, \, \mathbf{x}_t^{(1:m)}, \mathbf{x}_{t-1}^{(k)}} \left[ \log \frac{p_\theta(\mathbf{x}_{t-1}^{(k)} \mid \mathbf{x}_t^{(k)}, \mathbf{c})}{p_{\text{ref}}(\mathbf{x}_{t-1}^{(k)} \mid \mathbf{x}_t^{(k)}, \mathbf{c})} \right] \right) \right],$$
$$\tag{11}$$

where $\mathbf{x}_{1:T}^{(j)} \sim q(\mathbf{x}_{1:T} \mid \mathbf{x}_0^{(j)}, \mathbf{c})$, $\mathbf{x}_t^{(j)} \sim q(\mathbf{x}_t \mid \mathbf{x}_0^{(j)})$ and $\mathbf{x}_{t-1}^{(j)} \sim q(\mathbf{x}_{t-1} \mid \mathbf{x}_t^{(j)}, \mathbf{x}_0^{(j)})$ for $j = 1, \ldots, m$. Since the stagewise negative Plackett–Luce term $-\sum_j \left[ s_j - \log \sum_{k \geq j} \exp(s_k) \right]$ is convex in the score vector $s$, by Jensen's inequality we can push $\mathbb{E}_{t, \, \mathbf{x}_t^{(1:m)}}$ outside and obtain an upper bound:

$$L_{\text{approx}}(\theta) \leq - \mathbb{E}_{t, \, \mathbf{x}_t^{(1:m)}} \sum_{j=1}^{m} \left[ \beta T \, \mathbb{E}_{\mathbf{x}_{t-1}^{(j)}} \left[ \log \frac{p_\theta(\mathbf{x}_{t-1}^{(j)} \mid \mathbf{x}_t^{(j)}, \mathbf{c})}{p_{\text{ref}}(\mathbf{x}_{t-1}^{(j)} \mid \mathbf{x}_t^{(j)}, \mathbf{c})} \right] \right.$$
$$\left. - \log \sum_{k=j}^{m} \exp\left( \beta T \, \mathbb{E}_{\mathbf{x}_{t-1}^{(k)}} \left[ \log \frac{p_\theta(\mathbf{x}_{t-1}^{(k)} \mid \mathbf{x}_t^{(k)}, \mathbf{c})}{p_{\text{ref}}(\mathbf{x}_{t-1}^{(k)} \mid \mathbf{x}_t^{(k)}, \mathbf{c})} \right] \right) \right].$$

Using the standard reverse-Gaussian parameterization of the diffusion model, each inner expectation reduces to a KL-difference that, in turn, yields the denoising-error form with the usual timestep

weight $\omega(\lambda_t)$:

$$L_{\text{approx}}(\theta) = -\mathbb{E}_{\mathbf{c}, \mathbf{x}_0^{(1:m)}, t} \sum_{j=1}^{m} \left[ \beta T \omega(\lambda_t) \left( -\left\| \epsilon^{(j)} - \epsilon_\theta(\mathbf{x}_t^{(j)}, \mathbf{c}, t) \right\|_2^2 + \left\| \epsilon^{(j)} - \epsilon_{\text{ref}}(\mathbf{x}_t^{(j)}, \mathbf{c}, t) \right\|_2^2 \right) \right.$$

$$\left. - \log \sum_{k=j}^{m} \exp \left( \beta T \omega(\lambda_t) \left( -\left\| \epsilon^{(k)} - \epsilon_\theta(\mathbf{x}_t^{(k)}, \mathbf{c}, t) \right\|_2^2 + \left\| \epsilon^{(k)} - \epsilon_{\text{ref}}(\mathbf{x}_t^{(k)}, \mathbf{c}, t) \right\|_2^2 \right) \right) \right],$$

where $\epsilon^{(j)} \sim \mathcal{N}(\mathbf{0}, \mathbf{I})$, $\mathbf{x}_t^{(j)} = \sqrt{\bar{\alpha}_t}\, \mathbf{x}_0^{(j)} + \sqrt{1 - \bar{\alpha}_t}\, \epsilon^{(j)}$, and $\lambda_t = \alpha_t^2/\sigma_t^2$.

## B.2 Comparison With Existing Rank Preference Optimization Methods.

We found similar existing works that also aim to optimize at the list level. Defining the score as $\mathbf{s}^{(j)} = \|\epsilon^{(j)} - \epsilon_\theta(\mathbf{x}_t^{(j)}, \mathbf{c}, t)\|_2^2 - \|\epsilon^{(j)} - \epsilon_{\text{ref}}(\mathbf{x}_t^{(j)}, \mathbf{c}, t)\|_2^2$, Chen et al. (2025) derives a group pairwise loss (GPO) in the form of $L_{GPO} = \sum_{j=1}^{m}[(m - 2j + 1)\mathbf{s}^{(j)}]$ from the pairwise DPO loss, and introduces a standard rewards $\mathcal{A}_j = \frac{\mathbf{r} - \text{mean}(\mathbf{r})}{\text{std}(\mathbf{r})}$ for $\mathbf{r} = \{r^{(j)}\}_{j=1}^{m}$ representing the reward for each image. The reward is scored by some external evaluators. The image reward replaces the term $(m - 2j + 1)$, and the final objective becomes,

$$L_{\text{GPO}} = \sum_{j=1}^{m} [\mathcal{A}_j(\|\epsilon^{(j)} - \epsilon_\theta(\mathbf{x}_t^{(j)}, \mathbf{c}, t)\|_2^2 - \|\epsilon^{(j)} - \epsilon_{\text{ref}}(\mathbf{x}_t^{(j)}, \mathbf{c}, t)\|_2^2)].$$

Also, Karthik et al. (2024) proposes RankDPO, which uses lambda loss (Wang et al., 2018) for the list of images:

$$L_{\text{RankDPO}} = -\sum_{j=1}^{m} \sum_{k=j}^{m} \Delta_{j,k} \log \sigma(-\beta(\mathbf{s}^{(j)} - \mathbf{s}^{(k)})),$$

where $\Delta_{j,k} = |(2^{\phi(j)} - 1) - (2^{\phi(k)} - 1)| \cdot \left| \frac{1}{\log(1+\tau(j))} - \frac{1}{\log(1+\tau(k))} \right|$ represents the weight between image pairs $(\mathbf{x}^{(j)}, \mathbf{x}^{(k)})$, with $\phi(j), \phi(k)$ and $\tau(j), \tau(k)$ being the true scores and true ranks of $(\mathbf{x}^{(j)}, \mathbf{x}^{(k)})$. As the true scores are not available, Karthik et al. (2024) considers evaluation models such as HPS, Pick Score, Image Reward, etc., to compute the simulated scores for the images.

Different from existing methods, Diffusion-LPO does not require any additional scoring model to learn the ranked preference, as (a) the evaluator can cause extra computation cost, and (b) the automated responses from evaluators cannot replace the human annotation. As the goal is to learn user preferences, evaluator feedback may cause misleading information and contradict the actual human feedback.

## B.3 Detail Compariosn between GP-DPO and Duffusion-LPO

Let's consider the data point $\mathbf{x}^{(j)}$ in the preference ranking and its negative samples $\{\mathbf{x}^{(j+1)}, \cdots, \mathbf{x}^{(m)}\}$. When using the Group Pairwise DPO, the preference of data point $\mathbf{x}^{(k)}$ over its negative samples is reflected in the pairwise comparison:

$$\log \left( P_{\text{GP-DPO}}(\mathbf{x}^{(j)} > \mathbf{x}^{(k)}, \forall k \in \{j+1, \cdots, m\}) \right) = \log \left( \Pi_{k=j+1}^{m} \sigma(r(\mathbf{c}, \mathbf{x}^{(j)}) - r(\mathbf{c}, \mathbf{x}^{(k)})) \right).$$

For simplicity, we omit the expectation over $c$ and $\mathbf{x}_{0:T}^{(1:m)}$ in our analysis. With the notation we stated before, $s_\theta^{(i)} := \frac{p_\theta(\mathbf{x}_{0:T}^{(i)}|\mathbf{c})}{p_{\text{ref}}(\mathbf{x}_{0:T}^{(i)}|\mathbf{c})}$, we can find

$$\sum_{k=j+1}^{m} \log \sigma \left( \beta \log s^{(j)} - \beta \log s^{(k)} \right) = \sum_{k=j+1}^{m} \left[ \beta \log s^{(j)} - \log \left( (s^{(j)})^\beta + (s^{(k)})^\beta \right) \right]$$

$$\propto \beta \log s^{(j)} - \frac{1}{m-j+1} \sum_{k=j+1}^{m} \log \left( (s^{(j)})^\beta + (s^{(k)})^\beta \right).$$

Here, by scaling it with the scalar $1/(m-j+1)$, we can easily check the effect of pairwise preference optimization over a sub-list $\{\mathbf{x}^{(j)}, \cdots, \mathbf{x}^{(m)}\}$ in our setting.

For Diffusion-LPO in (3), we can find that the advantage of $\mathbf{x}^{(j)}$ over its negative samples is clearly characterized by a softmax function

$$\log\left(P_{\text{Diffusion-LPO}}(\mathbf{x}^{(j)} > \mathbf{x}^{(k)}, \forall k \in \{j+1, \cdots, m\})\right) = \beta \log s^{(j)} - \log \sum_{k=j}^{m} \exp\left(\beta \log s^{(k)}\right)$$

$$= \beta \log s^{(j)} - \log \sum_{k=j}^{m} (s^{(k)})^{\beta}.$$

Up to now, we have explicitly characterized how GP-DPO and Diffusion-LPO optimize the sub-list preference $\{\mathbf{x}^{(j)}, \cdots, \mathbf{x}^{(m)}\}$.

## C  EXPERIMENT SETTINGS

### C.1  EVALUATION DATA DETAILS

We provide the dataset details we used as follows:

**Pick-a-Pic dataset**  : Pick-a-Pic is a large-scale, publicly available dataset of human preferences for text-to-image generation (Kirstain et al., 2023). It is collected via a web application where users submit prompts, receive images from multiple diffusion backbones, and indicate their preferences between pairs of images. Version 1 (v1) contains over 500,000 examples, while the updated Version 2 (v2) extends this to more than one million. Each entry includes a prompt, two generated images, and a label for the preferred image (or a tie). In our experiments, as the Pick-a-Pic v2 is currently not available online, we use Pick-a-Pic v1 for training and its held-out test set for evaluation.

**Parti-Prompts**  : The Parti-Prompts dataset (Yu et al., 2022) was introduced to evaluate compositional and semantic capabilities of large-scale text-to-image models. It contains carefully curated prompts spanning diverse categories, such as objects, attributes, styles, and complex multi-object relations. The prompts are designed to systematically probe generation fidelity, compositionality, and visual grounding, making it a widely used benchmark for assessing the generalization of diffusion-based models. We adopt Parti-Prompts for standardized evaluation of text-to-image generation quality.

**HPSV2**  : The Human Preference Score v2 (HPS v2) dataset (Wu et al., 2023) is built from over 25,000 prompts and 98,000 images generated by Stable Diffusion, accompanied by 25,205 human preference annotations collected from the Stable Foundation Discord community. Each annotation compares multiple candidate images for the same prompt, with the user selecting a preferred image. This dataset was used to train a preference classifier fine-tuned from CLIP, which defines the HPS metric—a strong predictor of human judgments. We use the HPS v2 test set to benchmark how well our aligned diffusion models capture human preferences.

**InstructPix2Pix**  : InstructPix2Pix (Brooks et al., 2023) is a large-scale dataset for instruction-based image editing. It was synthetically generated by combining GPT-3 for producing editing instructions and Stable Diffusion with Prompt-to-Prompt for creating paired before/after images. The resulting dataset comprises over 450,000 examples of input images, natural language editing instructions, and corresponding edited outputs. Models trained on this dataset can generalize to real-world user-written instructions at inference time. In our evaluation, we use a standardized subset of 1,000 test samples to assess performance on instruction-guided image editing.

**ImgEdit**  : The ImgEdit-Bench dataset (Ye et al., 2025) is a comprehensive benchmark for instruction-based image editing. It contains 811 test cases across 14 sub-tasks, organized into three suites: a basic edit suite with nine common operations (add, remove, alter, replace,

style transfer, background change, motion change, hybrid edit, and cut-out), an Understanding–Grounding–Editing (UGE) suite that stresses spatial reasoning and fine-grained localization on challenging scenes, and a multi-turn suite that evaluates content understanding, content memory, and version backtracking. Each test case pairs a real image with natural-language editing instructions, and the benchmark defines multi-dimensional scores for instruction adherence, editing quality, detail preservation, and fake detection, computed using GPT-4o-based judgments and a forensic detector. In our experiments, we use only the ImgEdit-Bench evaluation data and follow its official test split and scoring protocol to assess image-editing performance.

## C.2 IMPLEMENTATION DETAILS

**Hyperparameters.** We adopt AdamW for training SD1.5 and Adafactor for training SDXL. The learning rate is set to $1 \times 10^{-8}$ with linear warmup, scaled by effective batch size. The global batch size is 2048: for pairwise methods, the effective batch contains 2048 pairs, and for listwise methods, it contains 2048 groups. For SD1.5, we set $\beta = 2000$ for Diffusion-DPO, Diffusion-LPO, and $\beta = 0.001$ for DSPO and DSPO-LPO. For SDXL, we set $\beta = 5000$ for Diffusion-DPO, Diffusion-LPO, and $\beta = 3000$ for DSPO and DSPO-LPO, following previous work.

## C.3 PERSONALIZED PREFERENCE IMAGE GENERATION

We introduce the pipeline of our personalized preference alignment tasks here. The setting is adapted from Dang et al. (2025).

**Personalized Preference Alignment** PPD is a framework designed to align diffusion models with individual-level human preferences. Personalization would lead to better generalization performance across different individuals (Yu et al., 2025). The pipeline operates in two stages. In the first stage, a vision–language model (VLM) summarizes each user's historical pairwise preference data into a dense user embedding. This embedding captures stylistic and semantic signals specific to the user, derived from only a few examples. In the second stage, the diffusion model is fine-tuned with these user embeddings as additional conditioning, injected via cross-attention layers. Training uses a variant of the Diffusion-DPO objective, which optimizes the model to generate images aligned with each user's preferences while maintaining regularization against a reference model.

**User Embedding Generation** Each user is represented using a small set of 4-shot preference pairs (caption, preferred image, dispreferred image). These examples are processed by the multimodal VLM LLaVA-OneVision (with a Qwen2 language backbone and multi-image capability). The final user embedding is obtained by extracting the hidden state of the last token of this profile from the Qwen2 encoder. A Chain-of-Thought (CoT) prompting is used: the VLM describes and compares each preference pair, then generates a textual user profile. The template of prompting can be found in Table11.

## C.4 COMPUTATIONAL OVERHEAD DISCUSSION

**Diffusion-LPO vs Diffusion-DPO** Diffusion-LPO can yield extra training time and GPU memory usage compared to Diffusion-DPO under one global step, and the additional overhead can increase as the list size increases. Table 5 provides the GPU memory usage and training time under different list size. Note that we set maximum list size to be 8 doesn't mean for all optimization steps we have to suffer from the longer training time. As 54% of the data in Diffusion-LPO also only have size 2, more than half of the time Diffusion-LPO are having same training time and memory cost as Diffusion-DPO.

Table 5: Training time and memory cost under different list size. All measurements are obtained on a single NVIDIA RTX A5000 GPU.

| List size | 2 | 3 | 4 | 5 | 6 | 7 | 8 |
|---|---|---|---|---|---|---|---|
| Time | 0.229s | 0.255s | 0.294s | 0.326s | 0.348s | 0.406s | 0.449s |
| Memory | 12815MB | 14065MB | 15285MB | 16542MB | 17753MB | 19007MB | 20234MB |

**Diffusion-LPO vs DP-GPO** Given a same global step for Diffusion-LPO and GP-DPO, the computation time for calculating the score of each image is the same. The main difference mainly comes from calculating the objective as single sum(as in Equation 4) or double sum(as in Equation 5). We measure the training time for one global step and the time difference is less than 1s, and the memory usage for both method is the same.

## D  ADDITIONAL QUALITATIVE ANALYSIS

### D.1  TEXT TO IMAGE QUALITATIVE RESULTS

We show more examples of images generated by Diffusion-LPO. Figure 6 provides examples for Diffusion-LPO over other baselines for SD1.5. From the top to the bottom, the prompts are:

- Beautiful girl;
- Woman Argentina;
- Realistic owl;
- mechanical bee flying in nature, electronics, motors, wires, buttons, lcd;
- beautiful, tranquil garden of love and peace;
- JEM doll 80s in a fur coat in the snow, thick outlines, bright colors, digital art, hard edges, detailed, anime style, dynamic pose, character design, art by sora kim, rinotuna, ilya kuvshinov;
- Duck.

Figure 7 shows more examples for Diffusion-LPO over other baselines for SDXL. From the top to the bottom, the prompts are:

- A blue airplane in a blue, cloudless sky;
- Portrait of an anime maid by Krenz Cushart, Alphonse Mucha, and Ilya Kuvshinov;
- Two Somali friends sitting and watching a Studio Ghibli movie;
- A jellyfish sleeping in a space station pod;
- A painting by Raffaello Sanzi portraying Kajol and symbiots Riot during the Renaissance era, showcased on Artstation;
- A person holding a very small slice on pizza between their fingers.

### D.2  IMAGE EDIT QUANTITATIVE AND QUALITATIVE RESULTS

We show more qualitative results for image editing in Figure 9. Edit prompts and original images are from the InstructPix2Pix dataset.

## E  GENERALIZATION FROM OTHER DPO FRAMEWORKS

The idea of switching from pairwise to listwise can be applied to most of the families of Diffusion DPO works. Here, we show one example of adapting Listwise preference into a DPO work, DSPO (Zhu et al., 2025). DSPO leverages score matching from human preference, and its original objective is,

$$\min_\theta \omega(t)||\nabla_{\mathbf{x}_t} \log p_\theta(\mathbf{x}_t|\mathbf{c}) - (\nabla_{\mathbf{x}_t} \log p(\mathbf{x}_t|\mathbf{c}) + \gamma \nabla_{\mathbf{x}_t} \log p_\theta(\mathbf{y}|\mathbf{x}_t, \mathbf{c}))||_2^2, \tag{12}$$

where $\omega(t)$ is the time-dependent function for score matching Song et al. (2021). DSPO model human preference with $p(\mathbf{y}|\mathbf{x}_t, \mathbf{c}) = p(\mathbf{x}_t \succ \mathbf{x}_t^l|\mathbf{x}_t^l, \mathbf{c}) = \sigma(r(\mathbf{c}, \mathbf{x}_t) - r(\mathbf{c}, \mathbf{x}_t^l))$ and reward calculated by $r(\mathbf{c}, \mathbf{x}_t) \propto -(||\epsilon_\theta(\mathbf{x}_{t+1}, t+1, \mathbf{c}) - \epsilon_{t+1}||_2^2 - ||\epsilon_{\text{ref}}(\mathbf{x}_{t+1}, t+1, \mathbf{c}) - \epsilon_{t+1}||_2^2)$. Plug into the objective(Equation 12), their loss function becomes,

$$\mathcal{L}_{\text{DSPO}} = A(t)||\epsilon_{\theta,t+1} - \epsilon_{t+1} - \lambda\gamma(1 - \sigma(r(\mathbf{c}, \mathbf{x}_t) - r(\mathbf{c}, \mathbf{x}_t^l))(\epsilon_{\theta,t+1} - \epsilon_{\text{ref},t+1})). \tag{13}$$

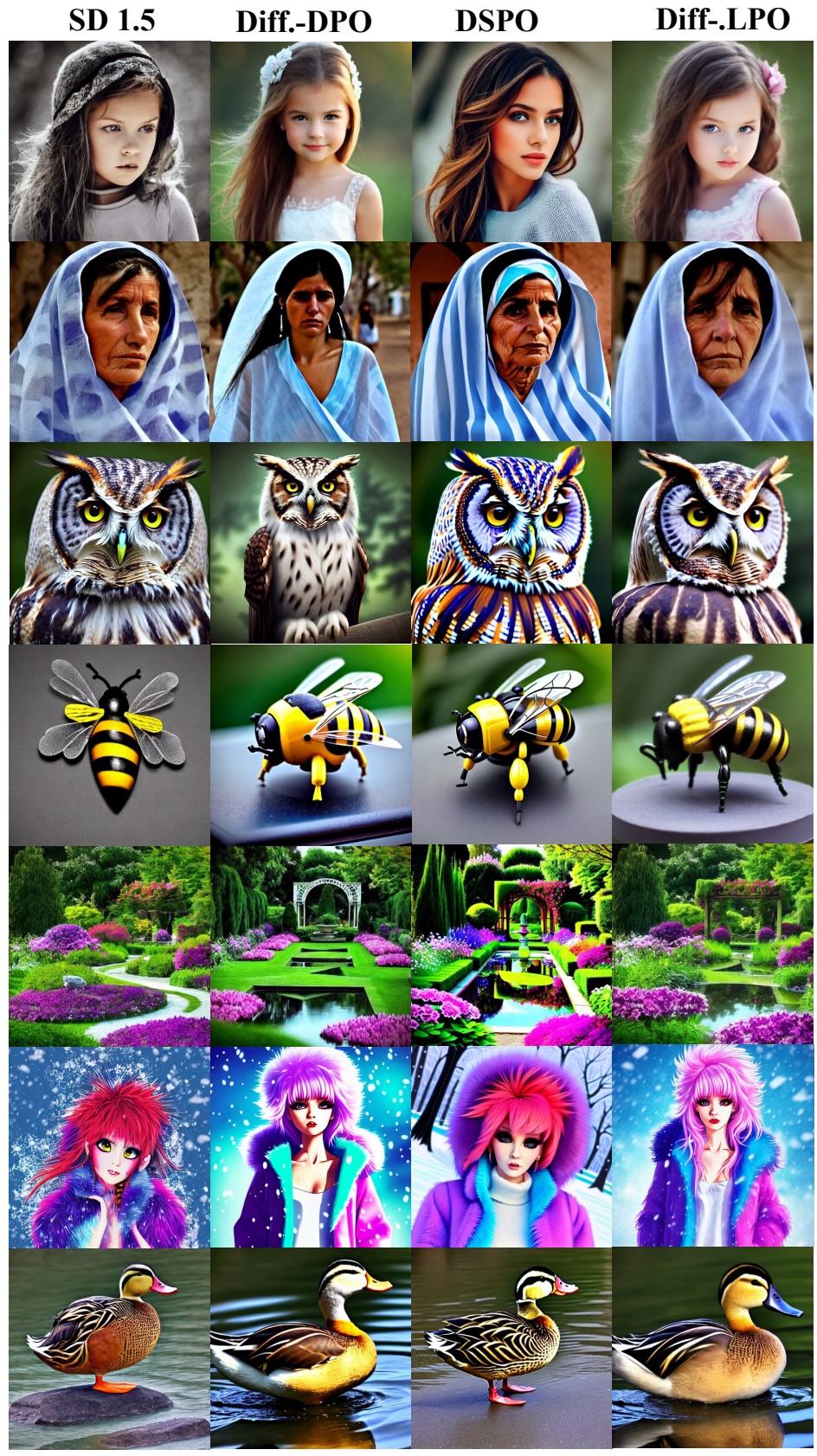

Figure 6: Qualitative illustrations for Diffusion-LPO over other baselines for SD1.5.

| SDXL | SFT | Diff.-DPO | DSPO | Diff-.LPO |
|---|---|---|---|---|

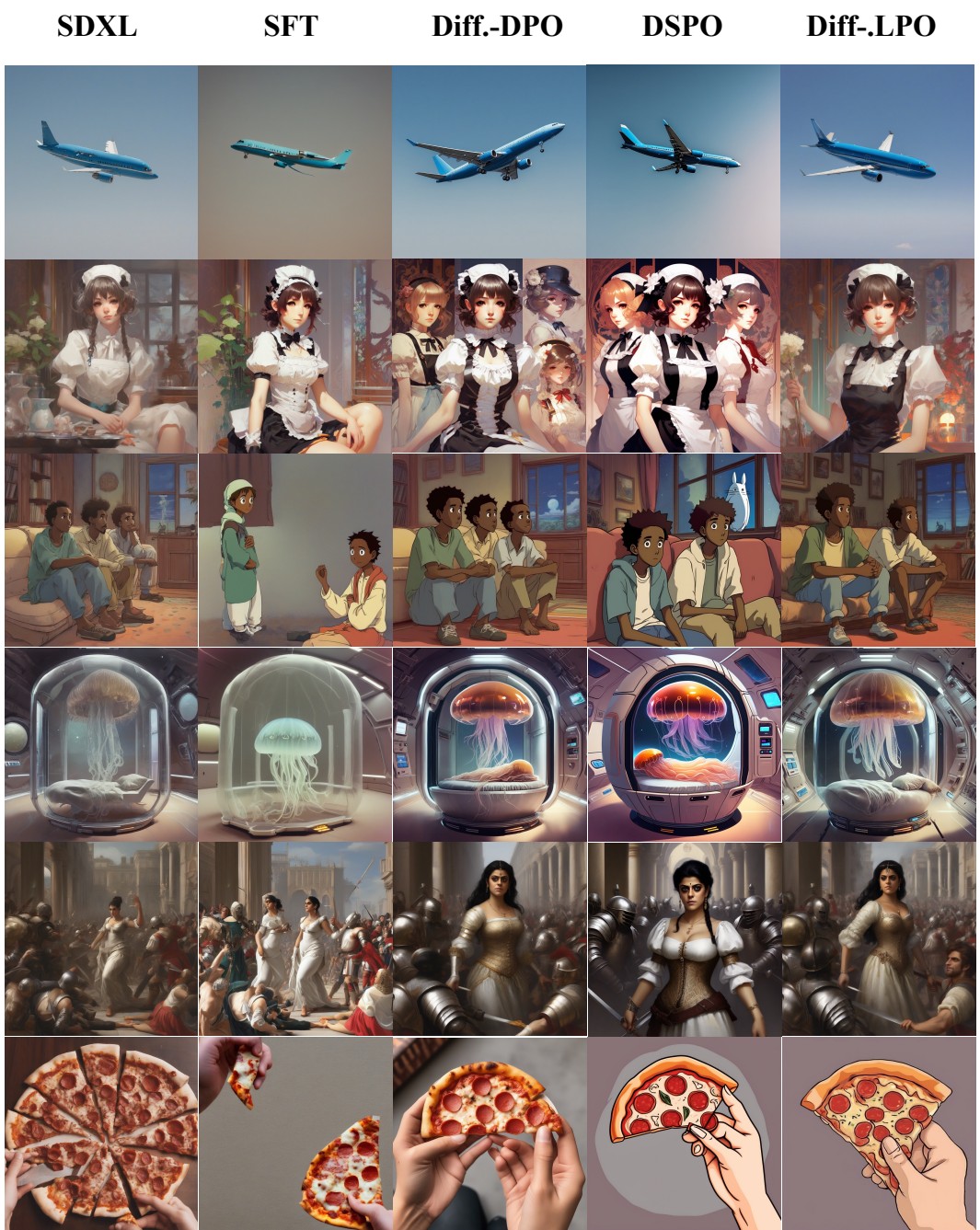

Figure 7: Qualitative illustrations for Diffusion-LPO over other baselines for SDXL.

| **Original** | **SD 1.5** | **Diff.-DPO** | **Diff.-LPO** |

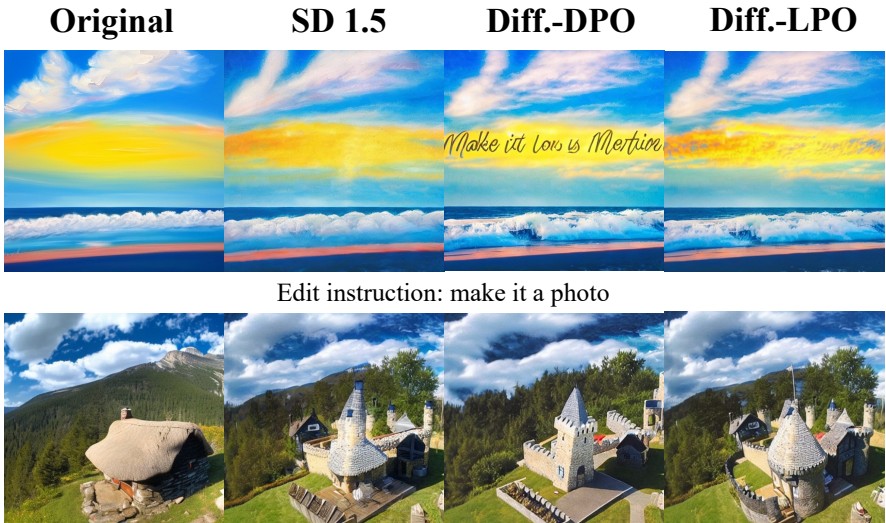

Edit instruction: make it a photo

Edit instruction: make the cottage a castle

Figure 8: For qualitative results, Diffusion-LPO faithfully follows the edit instructions while producing high-quality pictures. Diffusion-LPO produces more faithful and higher-quality edits. In the first example, only Diffusion-LPO successfully transforms a paint-like image into a realistic photograph, accurately reflecting the edit instruction. In the second example, while all methods modify the cottage into a castle as instructed, Diffusion-LPO generates a visually sharper and more coherent castle with realistic architectural details, underscoring its superior ability to handle fine-grained edits. More results are available in Figure 9.

To extend DSPO into listwise preference modeling, given a list of images with preferences $(\mathbf{x}^{(1)} \succ \mathbf{x}^{(2)} \succ \cdots \succ \mathbf{x}^{(m)})$ we can score-match the human preference through

$$p(\mathbf{y}|\mathbf{x}_t, \mathbf{c}) = p(\mathbf{x}_t^{(i)} \succ \mathbf{x}_t^{(i+1)} \succ ... \succ \mathbf{x}_t^{(m)}|\mathbf{x}_t^{(i+1)} \succ ... \succ \mathbf{x}_t^{(m)}, \mathbf{c}) = \frac{\exp\big(r(c, x^{(i)})\big)}{\sum_{k=i}^{m} \exp\big(r(c, x^{(k)})\big)}. \tag{14}$$

Therefore, a listwise preference optimization for DSPO(we denoted as DSPO-LPO) is,

$$\mathcal{L}_{\text{DSPO-LPO}} = A(t)||\epsilon_{\theta,t+1} - \epsilon_{t+1} - \lambda\gamma\Big(1 - \frac{\exp\big(r(c, x^{(i)})\big)}{\sum_{k=i}^{m} \exp\big(r(c, x^{(k)})\big)}\Big)(\epsilon_{\theta,t+1} - \epsilon_{\text{ref},t+1})). \tag{15}$$

In practice, we choose the highest rank image, $\mathbf{x}^{(1)}$, as the target of score matching ($i = 1$). A detailed derivation can be found at Appendix E.1.

### E.1 DSPO-LPO Objective

We start with the original objective of Equation 12. By modeling $p_\theta(\mathbf{y}|\mathbf{x}_t, \mathbf{c})$ as $\frac{\exp\big(r(\mathbf{c}, \mathbf{x}^{(i)})\big)}{\sum_{k=i}^{m} \exp\big(r(\mathbf{c}, \mathbf{x}^{(k)})\big)}$, we convert Equation 12 into:

$$\min_\theta \omega(t)||\nabla_{\mathbf{x}_t} \log \frac{p_\theta(\mathbf{x}_t|\mathbf{c})}{p(\mathbf{x}_t|\mathbf{c})} - \gamma\nabla_{\mathbf{x}_t} \log \frac{\exp\big(r(\mathbf{c}, \mathbf{x}^{(i)})\big)}{\sum_{k=i}^{m} \exp\big(r(\mathbf{c}, \mathbf{x}^{(k)})\big)}||_2^2 \tag{16}$$

$$= \min_\theta \omega(t)||\nabla_{\mathbf{x}_t} \log \frac{p_\theta(\mathbf{x}_t|\mathbf{c})}{p(\mathbf{x}_t|\mathbf{c})} - \gamma(1 - \frac{\exp\big(r(\mathbf{c}, \mathbf{x}^{(i)})\big)}{\sum_{k=i}^{m} \exp\big(r(\mathbf{c}, \mathbf{x}^{(k)})\big)})\nabla_{\mathbf{x}_t} r(\mathbf{c}, \mathbf{x}_t)||_2^2 \tag{17}$$

From the derivation in (Zhu et al., 2025), we can write the first term as:

$$\nabla_{\mathbf{x}_t} \log \frac{p_\theta(\mathbf{x}_t|\mathbf{c})}{p(\mathbf{x}_t|\mathbf{c})} \approx \epsilon_{\theta,t+1} - \epsilon_{t+1}, \tag{18}$$

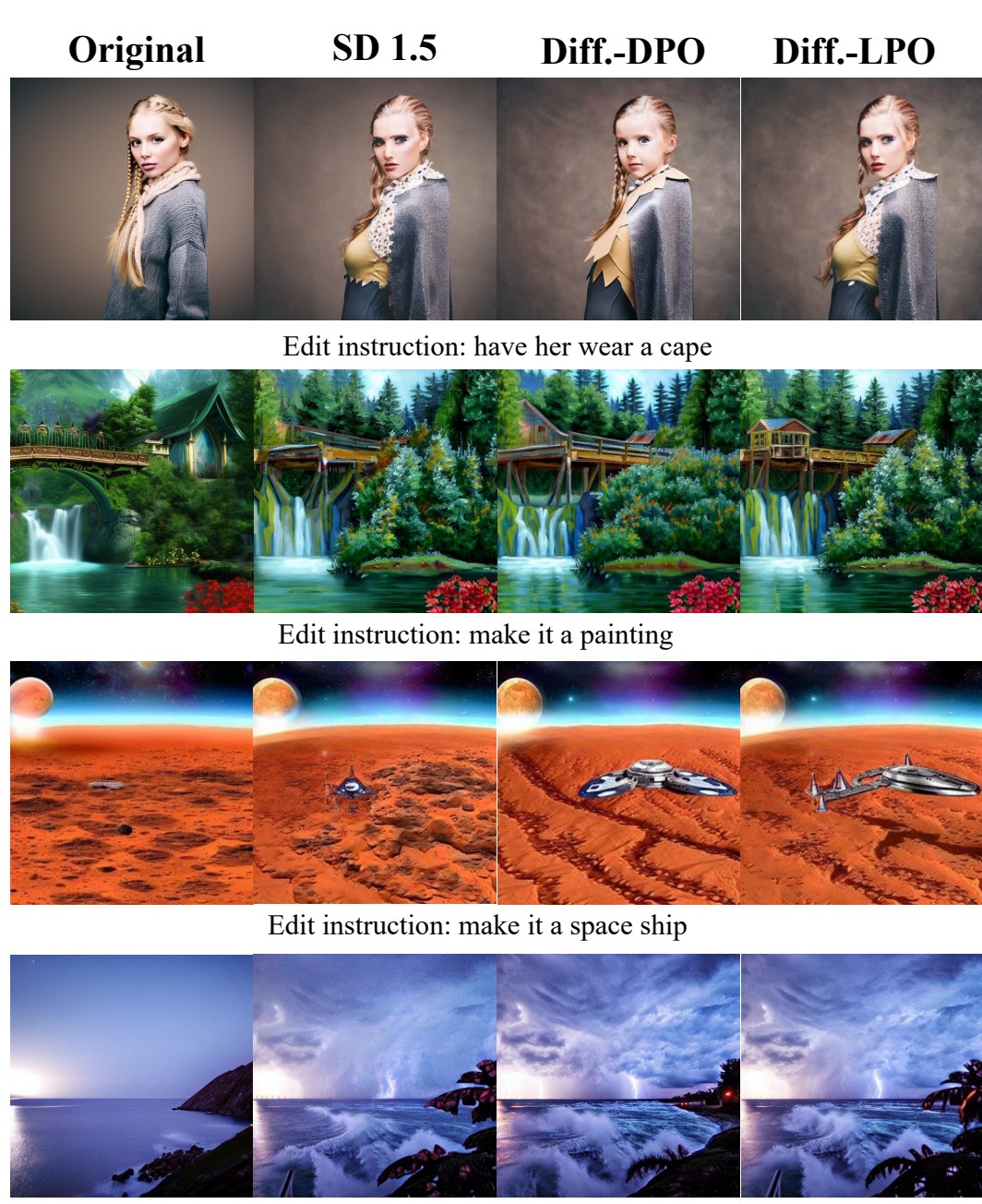

Figure 9: Qualitative results for image edit using SD1.5, finetuned over Diffusion-LPO and other baselines.

and that $\nabla_{\mathbf{x}_t} r(\mathbf{c}, \mathbf{x}_t)$ can be expressed in terms of the reference model:

$$\nabla_{\mathbf{x}_t} r(\mathbf{c}, \mathbf{x}_t) \approx \epsilon_{\theta,t+1} - \epsilon_{\text{ref},t+1}. \tag{19}$$

Thus the objective simplifies to:

$$\mathcal{L}_{\text{DSPO}} = A(t) \left\| \epsilon_{\theta,t+1} - \epsilon_{t+1} - \lambda\gamma\Big(1 - \sigma(r(\mathbf{c}, \mathbf{x}_t) - r(\mathbf{c}, \mathbf{x}^{(j)}))\Big)\big(\epsilon_{\theta,t+1} - \epsilon_{\text{ref},t+1}\big) \right\|_2^2, \tag{26}$$

where $A(t)$ absorbs the timestep-dependent weight and $\lambda$ is a constant factor arising from the Gaussian parameterization.

### E.2 TEXT-TO-IMAGE ALIGNMENT OF DSPO-LPO

We show the quantitative result of DSPO and DSPO-LPO in Table 6. We can observe that, by incorporating listwise preference optimization with DSPO, we further boost the performance across all metrics.

Table 6: Winrate results over original SDXL for Diffusion-LPO compared with DSPO, both fine-tuned on SDXL.

| Dataset | Method | PS↑ | HPS↑ | CLIP↑ | IM↑ | AES↑ |
|---------|--------|-----|------|-------|-----|------|
| Pick-a-Pic | DSPO | 59.2% | 77.4% | 56.6% | 62.9% | 45.9% |
| | DSPO+LPO | **77.1%** | **87.1%** | **64.0%** | **73.7%** | **51.5%** |
| Parti-Prompts | DSPO | 55.5% | 78.1% | 54.2% | 65.5% | 55.4% |
| | DSPO+LPO | **72.8%** | **82.9%** | **60.1%** | **73.1%** | **59.7%** |
| HPSV2 | DSPO | 58.4% | 77.3% | 47.8% | 63.7% | 48.2% |
| | DSPO+LPO | **74.6%** | **85.0%** | **56.3%** | **72.3%** | **51.9%** |

## F MORE RESULTS

### F.1 RESULTS ON DiT FAMILY MODELS

We provide additional experiments on SD3.5 (Esser et al., 2024), a DiT-based rectified-flow model. As shown in Table 7, Diffusion-LPO outperforms Diffusion-DPO across most cases.

Table 7: Winrate results over original SD3.5 for Diffusion-LPO compared with Diffusion-DPO.

| Dataset | Method | PS↑ | HPS↑ | CLIP↑ | IM↑ | AES↑ |
|---------|--------|-----|------|-------|-----|------|
| Pick-a-Pic | Diffusion-DPO | 54.3% | 54.6% | **53.5%** | 52.1% | 43.6% |
| | Diffusion-LPO | **58.6%** | **58.8%** | 51.4% | **53.6%** | **47.2%** |
| Parti-Prompts | Diffusion-DPO | 53.7% | 53.5% | **52.4%** | 50.9% | 47.9% |
| | Diffusion-LPO | **57.3%** | **57.7%** | 51.3% | **54.0%** | **49.2%** |
| HPSV2 | Diffusion-DPO | 52.7% | 54.6% | 51.1% | 52.2% | 44.3% |
| | Diffusion-LPO | **61.2%** | **58.8%** | **51.3%** | **55.5%** | **51.1%** |

**Discussion over online RL methods.** We also test FlowGRPO (Liu et al., 2025a) on the text to image alignment metrics. The result is in Table 8. RL-based methods such as FlowGRPO operate in a fundamentally different setting from ours: they perform online optimization using the reward model during training, whereas our method is fully offline and does not require any annotator or evaluator to score generated images in the training loop. For fairness, we additionally report comparisons using the publicly available FlowGRPO checkpoint. We observe that, FlowGRPO adopting PickScore as the reward model significantly outperforms the one using GenEval s the reward model, which heavily strengthens the choice of external evaluator in terms of helping the alignment. Our method, however, doesn't rely on any external evaluator.

### F.2 SIGNIFICANT TEST OF DIFFUSION-LPO VS GP-DPO

We provide further significant tests over the result in Table 2.

Table 8: Winrate results over original SD3.5 for FlowGRPO, with PickScore and GenEval as the reward model.

| Method | PS↑ | HPS ↑ | CLIP ↑ | IM ↑ | AES ↑ |
|---|---|---|---|---|---|
| SD3.5-FlowGRPO(PickScore) | 96.6% | 77.7% | 43.5% | 77.7% | 86.3% |
| SD3.5-FlowGRPO(GenEval) | 27.0% | 13.8% | 36.5% | 41.0% | 36.5% |

Table 9: P-values for Diffusion-LPO on SD1.5 in comparison to SD1.5 trained under GP-DPO.

| | PS | HPS | CLIP | IM | AES |
|---|---|---|---|---|---|
| Pick-a-Pic | 0.0065 | 0.0204 | 0.1923 | 0.2722 | 0.0008 |
| HPSv2 | 0.0056 | 0.0019 | 0.2467 | 0.0110 | 0.8112 |
| PartiPrompts | 0.0054 | 0.0082 | 0.1067 | 0.0097 | 0.0129 |

# G  THE USE OF LARGE LANGUAGE MODELS

The paper is primarily written by the authors, with only minor assistance from large language models (LLMs). In particular, LLMs were used for light editing tasks such as polishing grammar, refining phrasing, and suggesting alternative wordings. All technical content, derivations, and experimental design were conceived and carried out by the authors. No LLMs were used for generating research ideas, implementing methods, or conducting experiments.

# H  ETHICS STATEMENTS

We build on publicly available datasets and models, following established practices in prior work (Wallace et al., 2024; Podell et al., 2023; Rombach et al., 2022; Kirstain et al., 2023). All models are obtained from official repositories such as HuggingFace, and we respect associated licenses. While the Pick-a-Pic dataset may contain potentially unsafe content (e.g., depictions of violence or nudity), our use is strictly limited to research purposes within the dataset's intended scope. No additional human annotation was conducted in this study.

We are mindful of the broader societal impacts of generative models. Our work is intended to advance alignment techniques so that diffusion models better capture user preferences in a transparent and responsible manner. We do not release user-identifiable data and take care to avoid privacy risks. We also emphasize that preference alignment should be deployed in ways that respect diversity, minimize potential harm, and promote accessibility. All experiments were conducted in accordance with ethical and legal standards, with a commitment to openness, reproducibility, and integrity in reporting results.

# I  REPRODUCIBILITY STATEMENT

We use 8 H100-80G for our experiments, with GPU hours around 50 hours for finetuning SDXL for each experiment. We will release the model checkpoint and code source in the future.

# J  LIMITATION

In this work, the maximum list size is restricted to 8. We believe further exploration with larger lists and a systematic study of how list size influences optimization can provide additional insights. Moreover, evaluating Diffusion-LPO on a broader range of preference datasets would help further verify its effectiveness and generality.

Table 10: Confidence intervals for Diffusion-LPO on SD1.5 in comparison to SD1.5 trained under GP-DPO.

|  | PS | HPS | CLIP | IM | AES |
|---|---|---|---|---|---|
| Pick-a-Pic | [51.1%, 53.5%] | [50.3%, 52.3%] | [49.2%, 52.0%] | [48.1%, 53.5%] | [52.6%, 54.8%] |
| HPSv2 | [50.9%, 53.9%] | [52.2%, 55.8%] | [48.4%, 52.8%] | [50.3%, 52.3%] | [46.6%, 51.6%] |
| PartiPrompts | [50.9%, 53.9%] | [50.9%, 55.1%] | [49.3%, 52.3%] | [50.3%, 52.5%] | [50.6%, 53.6%] |

Table 11: Instructions for Embedding Generation and User Preference Prediction

| |
|---|
| **System Prompt** |
| You are an expert in image aesthetics and have been asked to predict which image a user would prefer based on the examples provided. |
| **COT Assistant Prompt** |
| You will be shown a few examples of preferred and dispreferred images that a user has labeled.

Here is Pair 1:
Here is the caption: [Caption for Pair 1]
Here is Image 1: [Image 1]
Here is Image 2: [Image 2]
Prediction of user preference: [1 or 2]

[...]

Here is Pair 4:
Here is the caption: [Caption for Pair 1]
Here is Image 1: [Image 1]
Here is Image 2: [Image 2]
Prediction of user preference: [1 or 2]

1. Describe each image in terms of style, visual quality, and image aesthetics.
2. Explain the differences between the two images in terms of style, visual quality, and image aesthetics.
3. After you have described all of the images, summarize the differences between the preferred and dispreferred images into a user profile.

Format your response as follows for the four pairs of images:

Pair 1:
Image 1: [Description]
Image 2: [Description]
Differences: [Description]

[...]

Pair 4:
Image 1: [Description]
Image 2: [Description]
Differences: [Description]

User Profile: [Description] |
| **Additional Assistant Prompt for User Preference Prediction** |
| Finally, you are provided with a new pair of images, unlabeled by the user. Your task is to predict which image the user would prefer based on the previous examples you have seen.
Format your response as follows:
Prediction of user preference: [1 or 2] |

