# OpenReview forum: "Towards Better Optimization For Listwise Preference in Diffusion Models"
_ICLR.cc/2026/Conference — ICLR 2026 Poster_

### Official Review · Reviewer_n3m2 · 2025-10-31

**Soundness:** 3
**Presentation:** 3
**Contribution:** 2
**Rating:** 6
**Confidence:** 3

**Summary:**

The paper proposes an extension of DiffusionDPO from pairwise preference to listwise preference where the elements within the list is ranked. The main idea applies Placket-Luce model instead of the usual Bradley Terry model for reward construction. The authors show the advantage of DiffusionLPO over naive extension of DPO such as GP-DPO and show noticeable gains in performance over other baselines on classical datasets and SD model series.

**Strengths:**

- The presentation is clear and the idea is straightforward.
- The theoretical justification that naive extension of DPO overestimates the positive/negative gap is illuminating.
- The results show good improvements over baselines over multiple standard datasets.

**Weaknesses:**

- The baseline comparisons are not sufficient. There many RL-based methods such as FlowGRPO that also optimize for reward signals that the authors miss. How does LPO compare with these online RL techniques?
- The optimized models are quite outdated (e.g. SD1.5 and SDXL). How does the method perform on newer models such as FLUX?

**Questions:**

I would like the author to address above questions.

---

> ### Author Response · Authors · 2025-11-21
>
> We thank the reviewer for their valuable feedback. We respond the reviewer’s comments in the following two replies.
>
> ---
>
> ## 1. Comparison with Online RL-based methods
>
> > "The baseline comparisons are not sufficient. There are many RL-based methods such as FlowGRPO that also optimize for reward signals that the authors miss. How does LPO compare with these online RL techniques?"
>
> RL-based methods such as FlowGRPO operate in a fundamentally different setting from ours: they perform online optimization using the reward model during training, whereas **our method is fully offline and does not require any annotator or evaluator to score generated images in the training loop**. FlowGRPO optimizes an absolute reward signal provided by a learned reward model, while DPO-style methods, including Diffusion-LPO, **only consume relative human preferences (pairs or lists) collected offline**. Because of this difference, **directly comparing our method to online RL approaches is not entirely apples-to-apples**: online RL can in principle leverage substantially more feedback from the reward model, at the cost of additional engineering complexity, computation, and a heavy dependence on the choice and quality of that reward model. In contrast, our goal in this work is to study how to better exploit offline listwise preference data without assuming access to a strong reward model during training.
>
> For fairness, we additionally report comparisons using the publicly available FlowGRPO SD3.5 medium checkpoints (see Table 1 below and Table 8 in the updated paper). We choose two checkpoints, one that uses PickScore as the reward model and the other that uses GenEval as the reward model.  As PickScore is trained to reflect human preference, the FlowGRPO result trained under PickScore achieves very high winrate over the original SD3.5-Medium baseline, while FlowGRPO trained under GenEval even underperforms the original SD-3.5 medium model. This shark contrast shows **the choice of reward model can be a decisive factor for online RL methods**.
> Since Diffusion-LPO, like other DPO-style approaches, only relies on offline human ranking data and does not use any external reward model during training,**it targets a different and often more practical regime where such evaluators are not available**. We therefore view FlowGRPO-style online RL as complementary rather than directly comparable to our offline listwise preference optimization setting.
>
> **Table 1.** Winrate results over original SD3.5 for FlowGRPO, with PickScore and GenEval as the reward model.
> | Method                    | PS ↑  | HPS ↑ | CLIP ↑ | IM ↑  | AES ↑ |
> |---------------------------|-------|-------|--------|-------|-------|
> | SD35-FlowGRPO (PickScore) | 96.6% | 77.7% | 43.5%  | 77.7% | 86.3% |
> | SD35-FlowGRPO (GenEval)   | 27.0% | 13.8% | 36.5%  | 41.0% | 36.5% |
>
> ---
>
> ## 2. Modern Rectified-flow Models in DiT Architecture
>
> > "The optimized models are quite outdated (e.g., SD1.5 and SDXL). How does the method perform on newer models?"
>
> We appreciate the reviewer’s recommendation to test modern rectified-flow architectures. Following this suggestion, we conducted experiments on SD3.5-medium[1], a DiT-based rectified-flow model. As shown in Table 2 (Also Table 7, Appendix F.1 in updated paper), LPO outperforms DPO across most cases. This demonstrates that Diffusion-LPO remains effective on contemporary DiT-based architectures.
>
> **Table 2.** Winrate results over original SD3.5 for Diffusion-LPO compared with Diffusion-DPO.
> | Method                        | PS   | HPS  | CLIP  | AES  | IM   |
> |-------------------------------|------|------|-------|------|------|
> | SD35-DPO (HPSV2)              | 52.7\% | 54.6\% | 51.2\% | 44.3\% | 52.2\% |
> | SD35-LPO (HPSV2)              | 61.2\% | 58.8\% | 51.3\%  | 51.1\% | 55.5\% |
> | SD35-DPO (Pickapic)           | 54.3\% | 54.7\% | 53.3\%  | 43.6\% | 52.1\% |
> | SD35-LPO (Pickapic)           | 58.6\% | 56.9\% | 51.4\%  | 47.2\% | 53.6\% |
> | SD35-DPO (PartiPrompts)       | 53.7\% | 53.5\% | 52.4\%  | 47.9\% | 50.9\% |
> | SD35-LPO (PartiPrompts)       | 57.3\% | 57.7\% | 51.3\%  | 49.2\% | 54.0\% |
>
> [1]: Esser, Patrick, et al. Scaling rectified flow transformers for high-resolution image synthesis. Forty-first international conference on machine learning. 2024.
>
> ---
>
> ## Closing Remark
>
> We sincerely appreciate the reviewer's valuable feedback. We hope our response addressed your questions and encourage you to favorably consider accepting the paper. Thank you once again for your support!

---

> > ### Comment · Reviewer_n3m2 · 2025-11-27
> >
> > I thank the author for addressing some of my concerns. I would like to keep my score.

---

### Official Review · Reviewer_Kumb · 2025-10-31

**Soundness:** 4
**Presentation:** 4
**Contribution:** 4
**Rating:** 8
**Confidence:** 5

**Summary:**

This paper proposes a novel algorithm to generalize the pair-wise preference learning to list-wise preferences, where a ranking of generations are available. The proposed method use a Plackett–Luce Model to directly model listwise preferences, as opposed converting one list preference to multiple pair-wise preferences like previous works. Through extensive experiments, the author show that the proposed method outperform baselines on T2I generation and image editing.

**Strengths:**

1. The proposed method is well-motivated and is built on the solid theoretical foundation of Plackett–Luce Model, as opposed to navie converting list preferences to pair-wise ones like GP-DPO.
2. The experiments are comprehensive, covering a wide range of tasks, datasets, metrics and base models. I'm fully convinced about the effectiveness of the proposed LPO based on the presented results.

**Weaknesses:**

1. I understand that the authors followed the base model choices of most literature in T2I preference learning and used SD 1.5 and SDXL as the base model. However, it would be better if the authors can demonstrate the effectiveness of proposed method on modern rectified-flow models in DiT architecture, such as Sana, Flux, SD3. This would make the contribution more relevant to the community.
2.  While I'm convinced of the effectiveness of Diffusion-LPO compared to baselines presented in Table 1, Table 2 is less impressive, with the win rate against GP-DPO around 50% (i.e random). The author should perform statistical significance analysis (i.e. confidence interval or p-value) to determine if these results are significant or indistinguishable from the null hypothesis (i.e. LPO and GP-DPO has same performance).
3. Pickscore and HPS in general are not a good metrics for image editing tasks. I suggest the author report more common metrics used in image editing. These includes conventional metrics like DINO, L1, CLIP [1].  Alternative, the author may choose to adopt a more "modern" image editing benchmarks that uses GPT4 as the judge model, such as ImageEdit,

[1] Zhang, Kai, et al. "Magicbrush: A manually annotated dataset for instruction-guided image editing." Advances in Neural Information Processing Systems 36 (2023): 31428-31449.
[2] Ye, Yang, et al. "Imgedit: A unified image editing dataset and benchmark." arXiv preprint arXiv:2505.20275 (2025).

**Questions:**

See weakness
Additional questions that did not affect my judgment:
What is the complexity of the proposed LPO and other methods listed in appendix B? are they both O(m^2) where m is the size of the list?

---

> ### Author Response · Authors · 2025-11-21
>
> We appreciate the reviewer's acknowledgment of our contribution! In the following replies, we address all the reviewer's comments point-by-point.
>
> ----
>
> ## 1. Modern Rectified-flow Models in DiT Architecture
>
> > "It would be better if the authors can demonstrate the effectiveness of proposed method on modern rectified-flow models in DiT architecture."
>
> We appreciate the reviewer’s recommendation to test modern rectified-flow architectures. Following this suggestion, we conducted experiments on SD3.5-medium[1], a DiT-based rectified-flow model. As shown in Table 1 (Also Table 7, Appendix F.1 in updated paper), LPO outperforms DPO across most cases. This demonstrates that Diffusion-LPO remains effective on contemporary DiT-based architectures.
>
> **Table 1.** Winrate results over original SD3.5 for Diffusion-LPO compared with Diffusion-DPO.
> | Method                        | PS   | HPS  | CLIP  | AES  | IM   |
> |-------------------------------|------|------|-------|------|------|
> | SD35-DPO (HPSV2)              | 52.7\% | 54.6\% | 51.2\% | 44.3\% | 52.2\% |
> | SD35-LPO (HPSV2)              | 61.2\% | 58.8\% | 51.3\%  | 51.1\% | 55.5\% |
> | SD35-DPO (Pickapic)           | 54.3\% | 54.7\% | 53.3\%  | 43.6\% | 52.1\% |
> | SD35-LPO (Pickapic)           | 58.6\% | 56.9\% | 51.4\%  | 47.2\% | 53.6\% |
> | SD35-DPO (PartiPrompts)       | 53.7\% | 53.5\% | 52.4\%  | 47.9\% | 50.9\% |
> | SD35-LPO (PartiPrompts)       | 57.3\% | 57.7\% | 51.3\%  | 49.2\% | 54.0\% |
>
> [1]: Scaling rectified flow transformers for high-resolution image synthesis. In Forty-first international conference on machine learning, 2024.
>
> ---
>
> ## 2. Statistical Significant Analysis
>
> > "The author should perform statistical significance analysis (i.e., confidence interval or p-value) to determine if these results are significant or indistinguishable from the null hypothesis (i.e. LPO and GP-DPO have the same performance)"
>
> Thank you for the insightful comments. Following your suggestion, we added standard deviations and performed statistical significance tests (confidence intervals and p-values), and the results in Table 2 and 3(also in Table 9, 10 in Appendix F.2 in updated paper) indicate the win rates of Diffusion-LPO vs GP-DPO.
>
> Overall, we find that Diffusion-LPO significantly outperforms GP-DPO on most metrics and datasets. In particular, PS and HPS are consistent with p-values < 0.05 across all three evaluation sets, and most of the AES/IM results also show confidence intervals whose lower bounds are above 50\%. By contrast, the CLIP metric is generally not statistically significant. These results indicate that, while CLIP does not reliably detect a difference between LPO and GP-DPO, most other metrics exhibit statistically significant gains, supporting that Diffusion-LPO provides a genuine performance improvement over GP-DPO rather than random variation.
>
> **Table 2.** p-value for Diffusion-LPO (SD1.5) vs SD1.5 trained under GP-DPO
> |                            | PS     | HPS    | CLIP   | AES    | IM     |
> |----------------------------|--------|--------|--------|--------|--------|
> | p-value(HPSV2)             | 0.0056 | 0.0019 | 0.2467 | 0.8122 | 0.0110 |
> | p-value(Pickapic)          | 0.0065 | 0.0204 | 0.1923 | 0.0008 | 0.2722 |
> | p-value(PartiPrompts)      | 0.0054 | 0.0082 | 0.1067 | 0.0097 | 0.0129 |
>
> **Table 3.** Confidence intervals for Diffusion-LPO (SD1.5) vs SD1.5 trained under GP-DPO.
>
> | Dataset      | PS                | HPS               | CLIP              | IM                | AES               |
> |-------------|-------------------|-------------------|-------------------|-------------------|-------------------|
> | Pick-a-Pic  | [51.1% , 53.5%]   | [50.3% , 52.3%]   | [49.2% , 52.0%]   | [48.1% , 53.5%]   | [52.6% , 54.8%]   |
> | HPSv2       | [50.9% , 53.9%]   | [52.2% , 55.8%]   | [48.4% , 52.8%]   | [50.3% , 52.3%]   | [46.6% , 51.6%]   |
> | PartiPrompts| [50.9% , 53.9%]   | [50.9% , 55.1%]   | [49.3% , 52.3%]   | [50.3% , 52.5%]   | [50.6% , 53.6%]   |

---

> ### Author Response · Authors · 2025-11-21
>
> ## 3. Metrics for Image Editing Tasks
>
> > "I suggest the author report more common metrics used in image editing. These include conventional metrics like DINO, L1, CLIP. Alternative, the author may choose to adopt a more "modern" image editing benchmarks that uses GPT4 as the judge model, such as ImageEdit"
>
> Thank you for the helpful suggestion! We now report more reasonable image-editing metrics and the GPT-evaluated benchmark(ImgEdit benchmark), as suggested by the reviewer. Table 4 includes results on the InstructPix2Pix dataset using DINO, L1, and CLIP as the metric. We can observe that both Diffusion-DPO and Diffusion-LPO improve substantially over the base model, and Diffusion-LPO further achieves consistently higher win rates than Diffusion-DPO across all three metrics.
>
> For the ImgEdit benchmark, we use GPT-4o as the based judge and directly compare the images generated from Diffusion-LPO with Diffusion-DPO. Under this setting, Diffusion-LPO achieves a win rate of 56.3\% over Diffusion-DPO. These results are included in Section 5.3 of the updated paper.
>
> **Table 4.** Winrate results over original SD1.5 on image editing tasks
> |        | Dino  | Clip  | L1    |
> |--------|-------|-------|-------|
> | Diffusion-DPO    | 70.4  | 72.4 | 85.8 |
> | Diffusion-LPO    | 74.7  | 76.0  | 89.6  |
>
> ----
>
> ## 4. The Complexity of the Diffusion-LPO and Other Methods
>
> > "What is the complexity of the proposed LPO and other methods listed in appendix B? are they both $O(m^2)$ where m is the size of the list?"
>
> For Diffusion-LPO, given a list of size $m$, we first compute a scalar score $r(c, x^{(j)})$ for each image $x^{(j)}$ in the list, which requires $O(m)$ computation. The listwise objective in Eq. (4) is then a single-sum Plackett–Luce loss, which can be implemented using cumulative log-sum-exp in $O(m)$ in the list length. The GPO method in [1] calculates the score of image in $O(m)$, but their method leverages external evaluators to calculate the standard reward $\mathcal{A}_j$ for each image $x^{(j)}$. This requires evaluating an extra reward model on all m images, which introduces additional overhead. RankDPO [2] conceptually optimizes a double-sum objective over $O(m^2)$ pairs, but the implementation can choose to pre-compute all the images scores $s^{(j)}$ in $O(m)$ and then the double sum calculates image scores as scalars. This quadratic scalar-loss computation is typically negligible compared to the cost of the backbone forward passes. However, RankDPO also requires true scores $\phi(j)$ for image $x^{(j)}$, which again may induce extra computation time for external evaluator to annotate each image.
>
> [1] Chen R, Lin W, Zhang Y, et al. Towards Self-Improvement of Diffusion Models via Group Preference Optimization. arXiv 2025.
>
> [2] Karthik S, Coskun H, Akata Z, et al. Scalable ranked preference optimization for text-to-image generation. Proceedings of the IEEE/CVF International Conference on Computer Vision 2025.
>
>
>
> ---
>
> ## Closing Remark
>
> We truly appreciate your valuable comments, which have contributed to the improvement of our paper. We hope that our responses demonstrate our commitment to addressing your comments. Thank you once again for your support!

---

> > ### Comment · Reviewer_Kumb · 2025-11-22
> >
> > Thanks the author for the response. My concerns have been addressed. I will keep my recommendation for acceptance.

---

> > > ### Author Response · Authors · 2025-11-22
> > >
> > > We’re grateful that our responses have addressed your concerns. Thank you so much for your support!

---

### Official Review · Reviewer_ATd9 · 2025-10-31

**Soundness:** 3
**Presentation:** 4
**Contribution:** 3
**Rating:** 6
**Confidence:** 3

**Summary:**

This paper proposes Diffusion-LPO, a listwise preference optimization framework for aligning text-to-image diffusion models with human preferences. Unlike prior Direct Preference Optimization (DPO) methods that rely solely on pairwise comparisons, the proposed method introduces Plackett–Luce–based listwise optimization, allowing consistent learning over the entire ranking of generated samples. The authors reconstruct the Pick-a-Pic dataset into ranked lists using DAG aggregation and demonstrate consistent improvements over Diffusion-DPO and DSPO on both SD1.5 and SDXL models, with +12% PickScore gain and improved HPSv2/CLIPScore.

**Strengths:**

1. The problem setting is clearly defined. The authors’ claim that human preferences are inherently based on ranked lists is well supported by the analysis of  Pick-a-Pic.

2. The paper provides a theoretical justification for why the Plackett–Luce model is more suitable than GP-DPO.

3.  A practical method is proposed for constructing listwise datasets from existing pairwise annotations.

**Weaknesses:**

1. There is no analysis of performance with respect to different list sizes.

2. The paper does not discuss the computational overhead introduced during training.

**Questions:**

1. Could the authors provide results when training with a fixed list size? I am curious about which list length contributes most effectively to performance improvement.

2. How do training time and GPU memory usage compare to GP-DPO? Since LPO introduces additional computation, please clarify how the training cost (in both time and memory) scales relative to the group-pairwise baseline.

3. The paper mentions that 54% of pair-wise data could be converted into list-wise form.
What kind of pairwise annotation design would be effective for increasing this percentage ? For example, are there particular annotation patterns or graph connectivity structures that facilitate more transitive ranking reconstruction?

---

> ### Author Response · Authors · 2025-11-21
>
> We thank the reviewer for the positive score and insightful feedback for our paper. In the following replies, we address all the reviewer's comments point-by-point.
>
> ---
>
> ## 1. Ablation study for the list size
>
> > "There is no analysis of performance with respect to different list sizes." "Could the authors provide results when training with a fixed list size?"
>
> Thank you for your comments! We realize that our original submission did not clearly highlight the choice of list size and have clarified this in the revised version. By design, our method does not enforce a single fixed list length; instead, in all main experiments we keep **all reconstructed lists with length of 8 as maximum**. This choice preserves the vast majority of available supervision while keeping training stable and efficient. Aggressive truncation or filtering can substantially reduce the effective signal of preference. If we require lists to have length exactly 8, only 23,087 valid lists remain, which is much smaller than the full set of over 400k lists.
>
> To assess the influence of the maximum list length, we compare performance under maximum list sizes of 4, 8, and 12 in Table 1 (also Table 3 in Section 6.2 in updated paper). The result shows that increasing the maximum list size from 4 to 8 brings 1\% of overall improvement, while increasing the list size from 8 to 12 yields only marginal differences. This suggests that **list size 8 for Diffusion-LPO in our setting is sufficiently large**, which is also consistent with the data distribution: around 95\% of the lists in our data have a list size less than or equal to 8.
>
> **Table 1.** Win rate of Diffusion-LPO (SD1.5) under different maximum list sizes on Pick-a-Pic test dataset.
>
> | Max List Size | Avg ↑ | PS ↑  | HPS ↑ | CLIP ↑ | IM ↑  | AES ↑ |
> |---------------|-------|-------|-------|--------|-------|-------|
> | m = 4         | 68.2% | 79.4% | 73.0% | 58.5%  | 68.5% | 61.6% |
> | m = 8         | 69.1% | 80.4% | 74.6% | 58.5%  | 68.1% | 64.0% |
> | m = 12        | 69.3% | 79.9% | 74.7% | 60.5%  | 69.2% | 62.1% |
>
> ----
>
> ## 2. Training time and GPU memory usage compared with DPO and GP-DPO
>
> > "The paper does not discuss the computational overhead introduced during training." "How do training time and GPU memory usage compare to GP-DPO?"
>
> Diffusion-LPO vs Diffusion-DPO: Diffusion-LPO can yield extra training time and GPU memory usage compared to Diffusion-DPO under one single step, and the additional overhead can increase as the list size increases. Table 2(also in Table 5 in Appendix C.4 in updated paper) provides the GPU memory usage and training time under different list sizes. All measurements are obtained on a single NVIDIA RTX A5000 GPU. Importantly, **setting the maximum list size to 8 does not mean that every optimization step incurs the cost of an 8-item list. As 54\% of the data in Diffusion-LPO also only have size 2, more than half of the updates have the same training time and memory cost as Diffusion-DPO**.
>
> Diffusion-LPO vs GP-DPO: Under matched conditions, the computation cost for Diffusion-LPO and GP-DPO is very similar. Given the same list, both methods calculate each image's score in the same way. The main difference mainly comes from how the loss is aggregated, i.e., calculating the objective as single sum(as in Equation 4) or double sum(as in Equation 5). We measure the training time for one global step(2,048 lists) and the time difference is less than 1s, and the memory usage for both method is the same.
>
> We add a discussion of the computational overhead of Diffusion-LPO and other methods in Appendix C.4 in the updated paper.
>
> **Table 2.** Training time and memory cost for processing one list under different list sizes.
> | List size | 2      | 3      | 4      | 5      | 6      | 7      | 8      |
> |-----------|--------|--------|--------|--------|--------|--------|--------|
> | Time      | 0.229s | 0.255s | 0.294s | 0.326s | 0.348s | 0.406s | 0.449s |
> | Memory    | 12815MB| 14065MB| 15285MB| 16542MB| 17753MB| 19007MB| 20234MB|

---

> ### Author Response · Authors · 2025-11-21
>
> ## 3. Pairwise Annotation Design
>
> > "The paper mentions that 54\% of pair-wise data could be converted into list-wise form. What kind of pairwise annotation design would be effective for increasing this percentage? "
>
> To increase the proportion of pairwise annotations that can be converted into listwise rankings, we believe two annotation patterns are particularly effective. First, offline datasets can be designed to encourage longer and more interconnected comparison chains.
> Instead of sampling pairs completely at random, one can prioritize pairs that share items with already annotated comparisons. This increases the connectivity of the comparison graph and leads to larger and more complete ranking DAGs.
> Second, adopting an online or interactive annotation protocol can further enhance listwise reconstruction: for example, once an annotator has indicated that $a>b$ and $c>b$, the system can proactively ask the annotator to compare $a$ and $c$, enabling the formation of a consistent triplet ranking. Such adaptive querying encourages transitivity and yields denser comparison graphs, ultimately increasing the fraction of data that can be aggregated into listwise structures.
>
> ---
>
> ## Closing Remark
>
> We sincerely hope that our responses have adequately addressed your comments. If you feel they have, we would be grateful if you could consider adjusting the evaluation of our manuscript accordingly. Thanks again for your valuable suggestions! We look forward to any further guidance you may have.

---

> > ### Comment · Reviewer_ATd9 · 2025-11-22
> >
> > Thank you for providing the additional experimental results.
> > Considering the performance improvements, I believe the computational overhead from increasing the list length is acceptable.
> > Moreover, I find your annotation idea extremely valuable.
> > Taking the practical applicability of this approach into account, I have decided to raise my score.

---

> > > ### Author Response · Authors · 2025-11-22
> > >
> > > Thank you very much for your positive feedback and appreciation of our work. We truly value your thoughtful comments and support!

---

### Official Review · Reviewer_FSKG · 2025-11-03

**Soundness:** 3
**Presentation:** 3
**Contribution:** 3
**Rating:** 6
**Confidence:** 4

**Summary:**

This paper proposes to align diffusion models with listwise preference. Instead of learning human preferences from ranking pairs, this paper learns human preferences from a ranked list. It extends the DPO objective under the Plackett-Luce model and proposes LPO, which is a framework that can be combined with any DPO-style training losses and achieve further gains. The authors construct a listwise ranking dataset by postprocessing the existing Pick-a-Pic dataset and extracting ranked lists. Experiments show that the proposed method is effective compared to baselines.

**Strengths:**

1. The idea of extending DPO to listwise preference data is novel. The use of the Plackett-Luce model to extend the Bradley-Terry model is well-motivated. The derivation is theoretically grounded, and the authors provide a clear connection to the diffusion processes.
2. The observation in the Pick-a-Pic dataset that a large portion of it can be aggregated into lists is important, allowing richer training signals without additional labeling costs.
3. Strong empirical results show consistent improvements over prior methods.

**Weaknesses:**

While the authors claim that LPO-style methods outperform prior DPO-style approaches, it is unclear whether the comparison is controlled for the same amount of data and training signal. For instance, are the models trained on the same set of image pairs or groups, and do they receive an equal number of optimization steps or preference annotations? If the listwise method effectively utilizes more comparisons per group (e.g., m(m−1)/2 relations in a list of size m), then it may benefit from a richer signal.

Additionally, the paper lacks a detailed analysis of why the improvement occurs. Is it primarily due to better preference supervision (e.g., higher-order transitivity), reduced noise, or more structured gradients? Such an analysis would significantly strengthen the empirical claims.

**Questions:**

see Weaknesses.

Is Diffusion-LPO implemented as an extension of Diffusion-DPO, DSPO, or both? The method is presented as a general framework compatible with arbitrary DPO-style objectives, but most results focus on Diffusion-DPO. Can the authors provide additional results showing that Diffusion-LPO also improves DSPO or other DPO variants, under matched conditions? Demonstrating consistent gains across multiple base algorithms would reinforce the method’s generality.

---

> ### Author Response · Authors · 2025-11-21
>
> We thank the reviewer for their insightful feedback and their recognition of our work's contribution. We respond the reviewer's comments in the following two replies.
>
> -------
>
> ## 1. Understanding the improvement
>
> > "it is unclear whether the comparison is controlled for the same amount of data and training signal. ... If the listwise method effectively utilizes more comparisons per group (e.g., m(m-1)/2 relations in a list of size m), then it may benefit from a richer signal" "Lack a detailed analysis of why the improvement occurs."
>
> The performance improvement primarily arises from two key factors. First, **the preference signal is enriched when we exploit listwise preference data instead of pairwise data**. We find that human preference data naturally forms directed acyclic graphs (DAGs) over image rankings. By reconstructing these DAGs into listwise ranking structures, our method benefits from richer and more coherent preference signal than pairwise annotations alone. Note that, in a single update step, Diffusion-LPO can leverage more structured preference relations from a list of size $m>2$, whereas standard DPO typically only sees a single pair. This "denser" signal per step is exactly the intended advantage of listwise modeling rather than an unfair use of extra data.
>
> Importantly,  the richer preference signal is constructed **under the same underlying human annotations**: as described in Appendix A, we reconstruct listwise groups purely by aggregating existing pairwise annotations in Pick-a-Pic into DAGs, without introducing any additional data or annotations.  Moreover, all methods are trained on the full dataset for 4 epochs on SD1.5 and 3 epochs on SDXL, ensuring they see the same annotations sufficiently. Thus, the gain does not come from more data or more training steps, but from more effective use of the same preference annotation.
>
> The second benefit comes from the fact that **Diffusion-LPO introduces a principled and efficient optimization objective that explicitly models the full preference ranking structure**. Given a preference list, Diffusion-LPO leverages the Plackett-Luce model for listwise preference learning, rather than sticking with Bradley-Terry model by extending the list into m(m-1)/2 pairs. We introduce the GP-DPO method to distangle these two source of benefit. GP-DPO has the objective $\sum_{j=1}^{m}\sum_{k=j+1}^{m}
> \log \sigma\left(r(c, x^{(j)}) -r(c,x^{(k)})\right)$, which extends the list into m(m-1)/2 pairs and performs pairwise DPO, but both GP-DPO and Diffusion-LPO use exactly the same listwise preference data and the same training hyperparameters. The only difference lies in how the listwise signal is encoded in the loss. We show the result of Diffusion-LPO vs GP-DPO in table 1 and Section 6.1 in the updated paper, and provide detailed statistical significance analysis (confidence intervals and p-values) in Tables 9 and 10 in the updated paper. From the result, we can observe that the optimization objective of Diffusion-LPO can generate better result than GP-DPO most of the time.
>
> Therefore, the gains of Diffusion-LPO stem from a data-driven reformulation that more faithfully captures human ranking structures, coupled with an optimization objective that exploit such structured preference information more effectively.
>
> **Table 1.** Win rate of Diffusion-LPO (SD1.5) vs SD1.5 trained under GP-DPO.
>
> | Dataset        | PS ↑              | HPS ↑             | CLIP ↑            | IM ↑              | AES ↑             |
> |---------------|-------------------|-------------------|-------------------|-------------------|-------------------|
> | Pick-a-Pic    | 52.3% ± 0.012     | 51.3% ± 0.010     | 50.6% ± 0.014     | 50.8% ± 0.027     | 53.7% ± 0.011     |
> | HPSv2         | 52.4% ± 0.012     | 54.0% ± 0.015     | 50.6% ± 0.018     | 51.3% ± 0.008     | 49.1% ± 0.020     |
> | Parti-Prompts | 52.4% ± 0.012     | 53.0% ± 0.017     | 50.8% ± 0.012     | 51.4% ± 0.012     | 52.1% ± 0.010     |

---

> ### Author Response · Authors · 2025-11-21
>
> ## 2. The difference in implementation
>
> > "Can the authors provide additional results showing that Diffusion-LPO also improves DSPO or other DPO variants, under matched conditions?"
>
> Thank you for your comments. We would like to clarify that Diffusion-LPO is not merely an extension of Diffusion-DPO or DSPO, but a **generalization from pairwise preference optimization to listwise preference optimization**. When the list size is 2, the list reduces to a pairwise comparison and the Diffusion-LPO objective degenerates exactly to Diffusion-DPO, so Diffusion-DPO is a special (and indeed the most common) case of Diffusion-LPO. As the reviewer correctly noted, our listwise optimization approach can incorporate arbitrary DPO-style objectives with a little additional effort of rewriting the objective once list-wise human preference structure exists. Specifically, as shown in Appendix E.2, we apply our listwise formulation to DSPO and obtain DSPO-LPO, where DSPO-LPO denotes the DSPO method with our listwise preference optimization objective. In these experiments, we keep all training hyperparameters ($\beta$, learning rate, number of epochs, etc) identical between DSPO and DSPO-LPO. Under these matched conditions, DSPO-LPO consistently outperforms DSPO, demonstrating that our listwise reformulation can also improve other DPO variants beyond Diffusion-DPO itself.
>
>
> ----
>
> ## Closing Remark
>
> We hope our responses can address your comments. Again, we would like to thank you for your support of our work and insightful comments!

---

> > ### Comment · Reviewer_FSKG · 2025-11-28
> >
> > I thank the authors for their response. I keep my score of acceptance.

---

### Author Response · Authors · 2025-12-03
**Revision Summary**

We appreciate the insightful and constructive feedback from the reviewers. We are encouraged that reviewers find our idea of extending DPO to handle listwise preference data to be well motivated and empirically effective. All four initial reviews recommended acceptance. Across the reviews, several strengths were consistently highlighted:

- Reviewer FSKG found our extension of DPO to listwise preference via the Plackett–Luce model to be novel, theoretically grounded, and well connected to diffusion processes;
- Reviewer ATd9 appreciated the clear problem setup and the justification for Plackett–Luce over GP-DPO;
- Reviewer Kumb emphasized the solid theoretical foundation and comprehensive experiments across tasks, datasets, metrics, and base models;
- Reviewer n3m2 found our analysis illuminating and noted that our method achieves good improvements over baselines on multiple standard datasets.

In response to the reviewers’ feedback, we have revised the manuscript accordingly and summarize the main updates below:

- **Significance of GP-DPO vs Diffusion-LPO**(Reviewer Kumb, FSKG): We added significance tests comparing Diffusion-LPO and GP-DPO, reporting confidence intervals and p-values in Tables 9 and 10 (Appendix F.2). Overall, Diffusion-LPO significantly outperforms GP-DPO on most metrics and datasets.
- **Diffusion-LPO on DiT family models**(Reviewer Kumb, n3m2): In response to requests for results on more modern DiT architectures, we added experiments on SD3.5-Medium (Table 7, Appendix F.1), showing that Diffusion-LPO remains effective on contemporary DiT-based models.
- **Computation cost vs Diffusion-DPO**(Reviewer ATd9): We additionally report the training time and GPU memory usage for Diffusion-DPO and Diffusion-LPO in Table 5 (Appendix C.4). While larger lists introduce some overhead, about 54% of our data have list size 2, so more than half of the updates have essentially the same cost as Diffusion-DPO.
- **Updated metric for image editing**(Reviewer Kumb): We revised our evaluation for image editing and now report DINO, L1, and CLIP as the main metrics and included the new results in Section 5.3. The results indicate that Diffusion-LPO provides better alignment for following editing instructions.
- **Ablation on maximum list size**(Reviewer ATd9): We added an ablation study on the maximum list size used by Diffusion-LPO, shown in Table 3 (Section 6.2). The results suggest that a maximum list size of 8 is sufficient in our setting.

All four reviewers responded positively to the rebuttal with recommendations towards acceptance. In particular, reviewer ATd9 raised their score from 6 to 8 after considering the new results and clarifications. We are grateful to the reviewers and Area Chairs for their careful evaluation and constructive suggestions.

---

### Meta-Review · Area_Chair_Bniu · 2026-01-06

**Summary:**

All reviewers acknowledge the novelty of the proposed method, its strong empirical performance, and its solid theoretical foundation, and the four initial reviews were uniformly positive. In response to reviewer feedback, the authors added statistical significance tests comparing Diffusion-LPO and GP-DPO, demonstrating the superior performance of Diffusion-LPO (Kumb, FSKG). They also provided additional experimental results on a modern DiT architecture (SD3.5-Medium) (Kumb, n3m2), an ablation study on the maximum list size (ATd9), runtime and memory analysis (ATd9), and expanded evaluation metrics (Kumb). These additions satisfactorily address the reviewers’ concerns. Overall, the paper is strong and well-executed, and we recommend acceptance.

**Reviewer Concerns:**

Based on the rebuttal and subsequent discussion, the Area Chair has verified that the authors have addressed all the concerns raised by the reviewers. Moreover, the paper received unanimously positive evaluations in the initial review phase. The questions from the reviewers are all addressed.

**Reviewer Scores:**

For this paper, the discussions between the authors and reviewers are smooth. All the reviewers have replied that their concerns have addressed. One of the reviewers (ATd9) even raise his score from borderline accept to a higher score before the score reverting.

---

### Decision · Program_Chairs · 2026-01-26

Accept (Poster)